# Bacterial origins of thymidylate metabolism in Asgard archaea and Eukarya

Jonathan Filée [1,4] ✉, Hubert F. Becker[2,3,4], Lucille Mellottee[2], Rima Zein Eddine[2], Zhihui Li[2], Wenlu Yin[2], Jean-Christophe Lambry [2], Ursula Liebl[2] & Hannu Myllykallio [2] ✉

Asgard archaea include the closest known archaeal relatives of eukaryotes. Here, we investigate the evolution and function of Asgard thymidylate synthases and other folate-dependent enzymes required for the biosynthesis of DNA, RNA, amino acids and vitamins, as well as syntrophic amino acid utilization. Phylogenies of Asgard folate-dependent enzymes are consistent with their horizontal transmission from various bacterial groups. We experimentally validate the functionality of thymidylate synthase ThyX of the cultured 'Candidatus Prometheoarchaeum syntrophicum'. The enzyme efficiently uses bacterial-like folates and is inhibited by mycobacterial ThyX inhibitors, even though the majority of experimentally tested archaea are known to use carbon carriers distinct from bacterial folates. Our phylogenetic analyses suggest that the eukaryotic thymidylate synthase, required for de novo DNA synthesis, is not closely related to archaeal enzymes and might have been transferred from bacteria to protoeukaryotes during eukaryogenesis. Altogether, our study suggests that the capacity of eukaryotic cells to duplicate their genetic material is a sum of archaeal (replisome) and bacterial (thymidylate synthase) characteristics. We also propose that recent prevalent lateral gene transfer from bacteria has markedly shaped the metabolism of Asgard archaea.

Canonical thymidylate synthase ThyA (EC 2.1.1.45) was once considered the only enzyme capable of catalyzing the de novo methylation of the essential DNA precursor dTMP (deoxythymidine 5′-monophosphate or thymidylate) from dUMP (deoxyuridine 5′-monophosphate). However, combined in silico, biochemical and genetic complementation tests led to the discovery of a new metabolic pathway that operates in the methylation of DNA precursors in numerous microbial species[1–3], relying on a novel thymidylate synthase, ThyX (EC 2.1.1.148). No sequence or structural homology exists between ThyA (found in ≈ 65% of microbial genomes) and ThyX [also known as Flavin dependent thymidylate synthase (FDTS)] flavoproteins (≈ 35%)[4]. In contrast to the homodimeric ThyA proteins, the active

site of ThyX flavoenzymes is located at the interface of three subunits of the homotetrameric protein complex and accommodates dUMP, NADPH, and the carbon donor methylene tetrahydrofolate ($CH_2H_4$folate, a vitamin B9 derivative)[5–7]. Consequently, formation of the ThyX tetramer is necessary for catalytic activity.

In the unique reductive methylation reaction catalyzed by ThyA, $CH_2H_4$folate functions both as a source of carbon ($C_1$-carrier) and reducing equivalents[8], thus leading to the formation of dihydrofolate ($H_2$folate) as the oxidation product of tetrahydrofolate ($H_4$folate) (Fig. 1a, left panel). $H_2$folate is subsequently reduced to $H_4$folate by dihydrofolate reductase (DHFR) FolA, as only reduced folate derivatives are functional in intermediary metabolism. Consequently, ThyA

[1]Évolution, Génomes, Comportement et Écologie, CNRS, IRD, Université Paris-Saclay, Gif-sur-Yvette, France. [2]Laboratoire d'Optique et Biosciences (CNRS UMR7645, INSERM U1182), Ecole Polytechnique, Institut polytechnique de Paris, F-91128 Palaiseau, France. [3]Sorbonne Université, Faculté des Sciences et Ingénierie, F-75005 Paris, France. [4]These authors contributed equally: Jonathan Filée, Hubert F. Becker. ✉e-mail: jonathan.filee@universite-paris-saclay.fr; hannu.myllykallio@polytechnique.edu

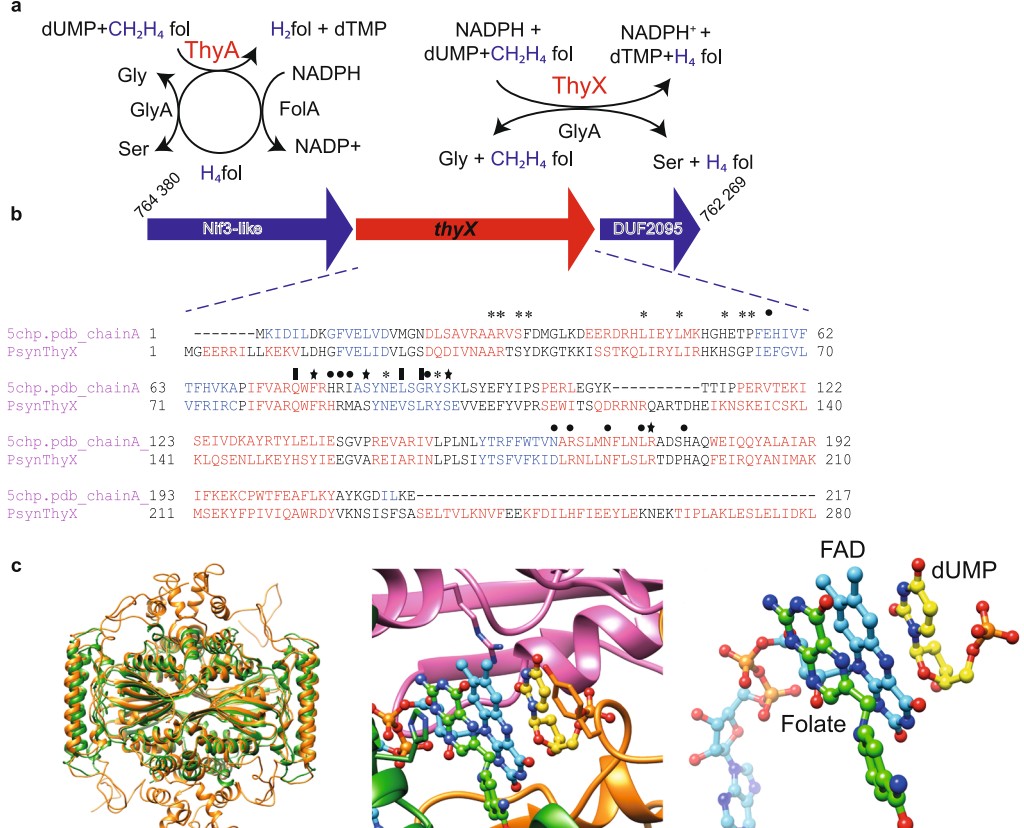

**Fig. 1 | ThyA- and ThyX-dependent folate cycles. a** ThyA catalyzes the methylation of dUMP to dTMP, leading to the formation of $H_2$folate that is subsequently reduced by FolA (left panel). The flavoenzyme ThyX uses methylene from $CH_2H_4$folate and acquires the reducing hydride from NADPH. $CH_2H_4$folate functions only as carbon source, resulting in $H_4$folate at the end of the catalytic cycle (right panel). Serine hydroxymethyltransferase GlyA is universally present. **b** *Psyn thyX* is located between the nucleotides 765,745 and 764,847 on the genome of *Psyn* (NZ_CP042905, 4 427 796 bp). The *Psyn thyX* gene is surrounded by the *nif3* and DUF2095 genes. For details, see text. Structure-based sequence alignment of *Psyn* ThyX with *Thermotoga maritima* ThyX (corresponding to PDB structure 5CHP) is also shown. Functionally important residues are indicated above the alignment. Asterisks refer to folate binding, stars to nucleotide-binding and filled circles to flavin binding residues. **c** Structural model for the *Psyn* ThyX homotetramer (left panel) using PDB structures 1O26, 3N0B and 6J61 as templates. Superposition of the model with PDB structure 4GT9 allowed the addition of FAD (cyan), dUMP (yellow), and folate (green) molecules (middle panel) to the structural model. The substrate and co-factor configuration is highlighted in the right panel.

and FolA form a functionally coupled adaptive unit embedded within the folate metabolism network[9]. $CH_2H_4$folate is then reformed in a reversible reaction by serine hydroxymethyltransferase (SHMT) GlyA using $H_4$folate and serine as substrates.

ThyX also uses $CH_2H_4$folate as $C_1$ carrier, but acquires the reducing hydride from NADPH and not from $H_4$folate. ThyX catalyzes dTMP formation using flavin-mediated hydride transfer from pyridine nucleotides, which is required for reduction of the methylene group, thus directly resulting in the formation of $H_4$folate (Fig. 1a, right panel). The ThyX reaction is substrate-inhibited[10], and catalytically less efficient than that used by ThyA enzymes[11], but maintains the folate in its reduced form (as $H_4$folate) at the end of the catalytic cycle. Consequently, *thyX*-containing organisms do not have an absolute requirement for FolA in their thymidylate metabolism (Fig. 1a), explaining the frequent absence of *folA* from *thyX*-containing organisms[12]. It remains unclear how two independent solutions for the synthesis of dTMP, the central building block of DNA evolved. Thymidylate synthases are known to participate in biosynthesis of nucleoside antibiotics and modified nucleotides in bacteriophages[12]. Phylogenetic analyses have revealed the existence of frequent and multiple lateral gene transfers during the evolution of ThyA and ThyX, resulting in their sporadic phylogenetic distribution mediated by non-homologous replacement[1,13]. ThyX appears overrepresented in genome-reduced and/or slow-growing prokaryotes[11,14], and changes in ThyX function of human microbiota associate with obesity-associated deficits in

inhibitory control towards responses to stimuli (e.g. food)[15]. Moreover, recent metagenomic read-mapping results indicate that *thyX* carrying bacteriophages are frequently found in oceans with distinct geographic distributions[16].

Despite the essential role of thymidylate synthase for DNA synthesis, details on the experimental characterization of archaeal thymidylate synthases remain scarce. While the halophilic archaeon *Haloferax volcanii* contains the gene for canonical thymidylate synthase ThyA, in the closely related halophilic archaeon *Halobacterium salinarum* ThyA homologs were not identified, whereas in *Hbt. salinarum* functional *thyX* was identified by genetic complementation[3]. Biochemically dTMP biosynthesis has been demonstrated in cell-free extracts of two different archaeal species[17,18]: *Methanosarcina thermophila* and *Sulfolobus solfataricus* (now referred to as *Saccharolobus solfataricus*). Although the corresponding enzymes were not purified or identified in this study, dTMP formation was detected from dUMP using externally added isotope-labeled formaldehyde and either chemically modified folates present in the cell extract (tetrahydrosarcinapterin; *M. thermophila*) or added synthetic fragments of sulfopterin (*S. solfataricus*). Indeed, biochemical studies have indicated the presence of at least six distinct, chemically modified, but thermodynamically analogous, $C_1$ carriers that function in archaeal central biosynthetic networks and/or energy-yielding reactions[19,20]. These chemical modifications differ among various archaeal species, partially explaining why the experimental characterization of archaeal

thymidylate synthases has not been followed up to date. Recent phylogenetic studies have suggested an archaeal origin for the energy-producing Wood-Ljungdahl pathway that is dependent on tetrahydromethanopterin (H4MPT), frequently found in methanogenic archaea[21]. This archaeal pathway may have contributed to the early origin of methanogenesis and the emergence of the use of reduced one-carbon compounds as carbon source (methylotrophy) in bacteria.

An archaeal superphylum called Asgard was recently discovered that includes the closest known archaeal relatives of eukaryotes[22-24]. This striking discovery has solidified the two-domain tree of life hypothesis in which Eukaryotes have emerged from within the archaeal tree[25]. Even if the true identity of the archaeal ancestors of Eukaryotes is still being debated, Asgard archaea occupy a pivotal position in the archaeal/eukaryotic phylogenetic trees[22,24]. Thus, studying the thymidylate and folate metabolism in the Asgard group is of particular interest to better understand the evolution of essential pathways for DNA, RNA, and protein synthesis, as well as their role in catabolic reactions. This is also underlined by the fact that pioneering genomic studies on Lokiarchaeota suggested the absence of biosynthetic capacity for tetrahydromethanopterin and tetrahydrofolate cofactors, but nevertheless the presence of some folate-dependent enzymes implicated in amino acid utilization[23].

In this study, we have exploited the recent increase in Asgard metagenome and genome sequence information to investigate thymidylate synthases and folate-dependent metabolic networks in more than 140 complete or nearly complete Asgard genomes. Our sequence similarity searches revealed a high level of similarity between the protein sequences of bacterial and Asgard folate-dependent enzymes, including thymidylate synthases. Our genomic and phylogenetic analyses suggested that Asgard archaea have 'hijacked' bacterial folate-dependent enzymes and pathways to support their metabolism. Experimental analyses of the archaeal thymidylate synthase revealed that ThyX from the recently cultured Asgard archaeon *Candidatus* Prometheoarchaeum syntrophicum strain MK-D1[23] (further referred to as *Psyn*) is functional as thymidylate synthase in bacterial cells and efficiently interacts with bacterial folates. Our combined experimental and computational data suggest that the patchy phylogenetic distribution and phylogenetic incongruences of functional folate-dependent enzymes from Asgard archaea reflect their independent horizontal transmission from various bacterial groups. These horizontal gene transfer (HGT) events have potentially rewired the key metabolic reactions in an important archaeal phylogenetic group. We also propose that the eukaryotic thymidylate synthase was transferred from bacteria to the protoeukaryotes during eukaryogenesis.

## Results

### Identification of a ThyX orthologue in *Candidatus* Prometheoarchaeum syntrophicum strain MK-D1 (*Psyn*)

As an initial approach to investigate the Asgard archaeal thymidylate and C$_1$ metabolism, we searched for *thyX* and *thyA* homologs in sequence databases for archaeal genomes or metagenome-assembled genomes (MAGs). These studies led to the identification of both *thyX* and *thyA* sequences in the genomes of the understudied and diverse group of Asgard archaea. Interestingly, our similarity searches identified a ThyX candidate in *Psyn* (Supplementary Fig. 1), which is up to 51.44% identical to bacterial ThyX sequences at the protein sequence level (*Zixibacteria*, e-value ≈10$^{-78}$ or *Calditrichaeota* bacterium, e-value ≈10$^{-76}$). The observed e-values are very low and are adjusted to the large sequence database size. Therefore, the quality of the observed hits is very high. As *Psyn* is the only known example of *Asgard* archaea that can be cultivated and its genome, lacking *thyA*, is completely sequenced, we concentrated our efforts on this archaeal ThyX.

Transcriptome analyses of RNA extracted from enriched cultures of *Psyn* indicated that *thyX* from this strain is expressed [with a Reads Per Kilobase of transcript per Million mapped reads (RPKM) value of

225.37 using the data set from the sequence read archive (SRA) DRR199588[23]], in agreement with previous analyses (see also Supplementary Fig. 2). This gene encodes a protein with a predicted molecular mass of 35, 294 Da. Its genomic environment (Fig. 1b and Supplementary Fig. 1) comprises an upstream gene coding for a Nif3-like protein with a length of 259 residues (29,026 Da) and a downstream gene encoding a domain of unknown function DUF2095. The physical association of *Psyn thyX* with the Nif3 family encoding gene is of interest, as this family of proteins may correspond to GTP cyclohydrolase 1 type 2, which converts GTP to dihydroneopterin triphosphate and may function in folic acid synthesis. The structure-based sequence alignment of *Psyn* ThyX with *Thermotoga maritima* ThyX (PDB structure 5CHP) predicts the conservation of functionally important residues involved in folate-, nucleotide- and flavin-binding (Fig. 1b). The marked degree of sequence similarity allowed the construction of a high-quality structural model for *Psyn* ThyX suggesting its functional significance as thymidylate synthase and nucleotide and folate binding protein. More specifically, we constructed a model of the ThyX homotetramer based upon PDB structures 1O26, 3N0B and 6J61 as templates and using the protein structure modeling program Modeller[26] (Fig. 1c, left panel). Structural superposition of the model with PDB structure 4GT9 using the Chimera software[27] allowed the addition of FAD, dUMP and folate molecules into the active site of the model (Fig. 1c, middle panel). Importantly, the model suggests the highly plausible transfer of 1C units via the N5 atom of the FAD cofactor to the accepting carbon of dUMP (Fig. 1c, right panel).

### *thyX* and *thyA* are present in Asgard archaea

As our sequence similarity searches suggested a close link between *Psyn* and bacterial ThyX sequences, we obtained more detailed insights into the thymidylate synthase gene distribution and phylogeny in Asgard archaea. In particular, in addition to the complete genome of cultivated *Psyn*, we analysed more than 140 MAGs of uncultivated Asgard archaea (Fig. 2). Our analyses revealed that *thyX* is present only in a small subset of lineages with a patchy (sporadic) gene distribution. We found *thyX* genes to be present in 33 out of a total of 141 Asgard (meta)genomes: 6/52 in the Loki lineage, 6/16 in Heimdall, 0/5 in Hel, 1/1 in Odin, 6/38 in Thor, 0/1 in Wukong, 1/2 in Borr, 1/12 in Hod, 3/3 in Gerd, 7/10 in Hermod, 0/1 in Kari and 2/2 in Baldr. In contrast, *thyA* genes are widely distributed among Asgard archaea: 93 out of a total of 141 (36/52 for Loki, 13/16 for Heimdall, 5/5 for Hel, 1/1 for Odin, 31/38 for Thor, 0/1 for Kari, 1/1 for Wukong, 1/2 for Borr, 3/12 for Hod, 0/3 for Gerd, 1/10 for Hermod, 1/2 for Baldr). Approximately twenty Asgard MAGs encode for both *thyX* and *thyA* genes, although *thyX* genes appear often partial or fragmented in this latter case. In some cases, both *thyX* and *thyA* appear to be absent, likely reflecting the fact that some of the analysed MAGs correspond to partially assembled genomes.

### Automated prediction of gene transfers between *Psyn* and bacteria

Considering the high level of sequence similarity of *Psyn* ThyX and bacterial orthologs, we investigated the possibility of HGT between bacteria and *Psyn* using HGTector. This automated method is well suited for predicting potential recent gene transfers[28]. The method performs a systematic analysis to detect genes that do not have a typical "vertical" history, thus detecting atypical genes that may correspond to recent HGTs.

Using HGTector, we predicted 149 HGT events [supplementary Data 1 (sheet "potential gene transfers")], including *Psyn thyX*, when all the ≈4000 *Psyn* predicted protein-coding genes were analysed (Fig. 3). In this plot, each dot represents one gene, and likely horizontally transferred genes are indicated in yellow. As depicted in Fig. 3, many *Psyn* folate-dependent genes (purple legends in Fig. 3a; a more detailed figure with annotations is available as supplementary Fig. 3) were also

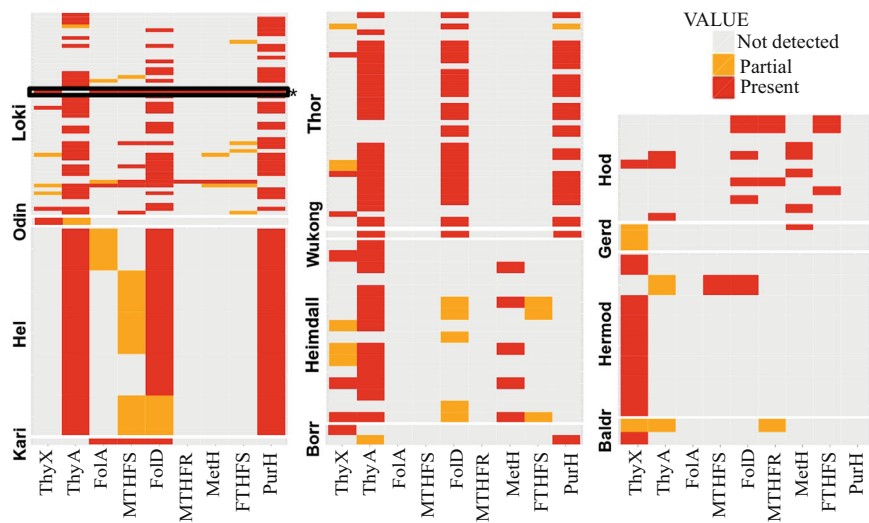

**Fig. 2 | Heatmap for phyletic distribution of different folate-dependent enzymes in Asgard archaea.** The red color indicates the presence of the entire gene, orange designates a partial copy. *Psyn* is indicated with an asterisk and its gene distribution is framed in black. The (meta)genomic assembly and gene names are in supplementary Data 1 (sheet "distribution of folate genes").

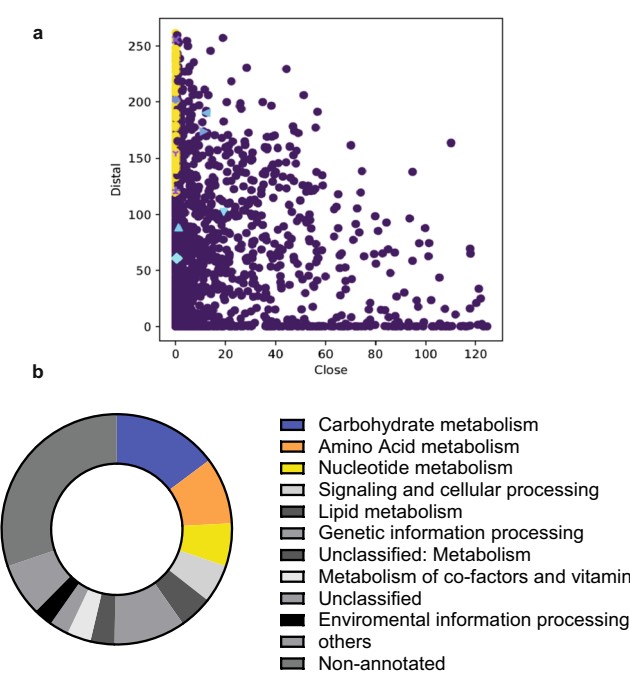

**Fig. 3 | Automated prediction of gene transfers between *Psyn* and bacteria.** **a** Plot indicating the distributions of "close" and "distal" scores revealed by similarity searches. The scatter plot was created with HGTector. Potentially transferred genes (*n* = 149) are indicated in yellow, whereas the other coloured labels refer to *Psyn* folate-dependent enzymes (for annotation, see supplementary Figs. 2 and 3). **b** Distribution of the functional groups for potentially transferred *Psyn* genes obtained using the BlastKOALA annotation[29].

potentially transferred from bacteria. The BlastKOALA annotation[29] of these candidate transfers indicates their functional distribution, particularly in amino acid, nucleotide, and carbohydrate metabolism (Fig. 3b and Supplementary Data 1). This is of interest considering the syntrophic amino acid utilization of *Psyn*[23].

### *thyX* was transferred from bacteria to diverse Asgard archaea

To better understand the origin and evolution of the thymidylate synthases ThyX and ThyA in Asgard archaea, we have reconstructed their phylogenies (Fig. 4). Note that Asgard sequences are indicated in red, other archaea in blue, bacteria in black and Eukarya in green in the obtained phylogenetic trees. The unrooted ThyX phylogeny supported with the bootstrap analyses (Fig. 4 and Supplementary Fig. 4 for a detailed version of the tree with the taxon names) shows that Asgard ThyX sequences are deeply polyphyletic; they are scattered into several monophyletic groups. *Psyn* ThyX (indicated by the asterisk) clusters along with some, but not all, Loki and Heimdall sequences, together with bacteria belonging to the *Deinococcus/Thermus* group. Additional Asgard sequences present in this tree cluster together with the different bacterial lineages. The only eukaryotic ThyX sequence from *Dictyostelium* branches with α proteobacteria, suggesting a transfer between bacteria and eukarya. This phylogenetic analysis supports the existence of multiple and independent HGT events of *thyX* from different bacteria into Asgard genomes. As control, the unrooted phylogeny for the evolutionary conserved MCM replicative helicase readily resolved eukaryotic, "archaeal" and Asgard groups, indicating that our methodology for tree constructions is robust (Fig. 4, middle inset "MCM helicase"). This latter result is also in agreement with the placements of the Asgard sequences in the B-family DNA Polymerase[30] and DNA Gyrase[31] phylogenies for which the Asgard form monophyletic groups with other archaeal sequences.

### ThyA may correspond to the ancestral thymidylate synthase in the analysed *Asgard* archaea

In contrast to ThyX, the ThyA phylogeny (Fig. 4 and supplementary Fig. 5 and on-line material for a detailed version of the tree with the taxon names) indicates that Asgard sequences group together with other archaeal sequences. The analyzed set of ThyA sequences of Asgard archaea are distributed into at least three clusters that appear related to diverse methanogenic archaea. This pattern is very different from the polyphyletic pattern of ThyX (Fig. 4), although we observe potential gene transfers between the different archaea (e.g. Hodarchaea and methanogens and Lokiarchaea and Helaarchaeota indicated in yellow in Fig. 4). The widespread presence of *thyA* in Asgard metagenomes (93/144 genomes), together with our phylogenetic analyses (Fig. 4), suggest the most parsimonious scenario where ThyA was present in the ancestor of this set of Asgard archaea. However, this tree topology is also compatible with the more complex scenarios of i) the Asgard ancestor inheriting its *thyA* gene from another archaea, followed by subsequent gene transfers to diverse methanogens, or ii) initial *thyA* transfers to methanogenic archaea and subsequent ones to

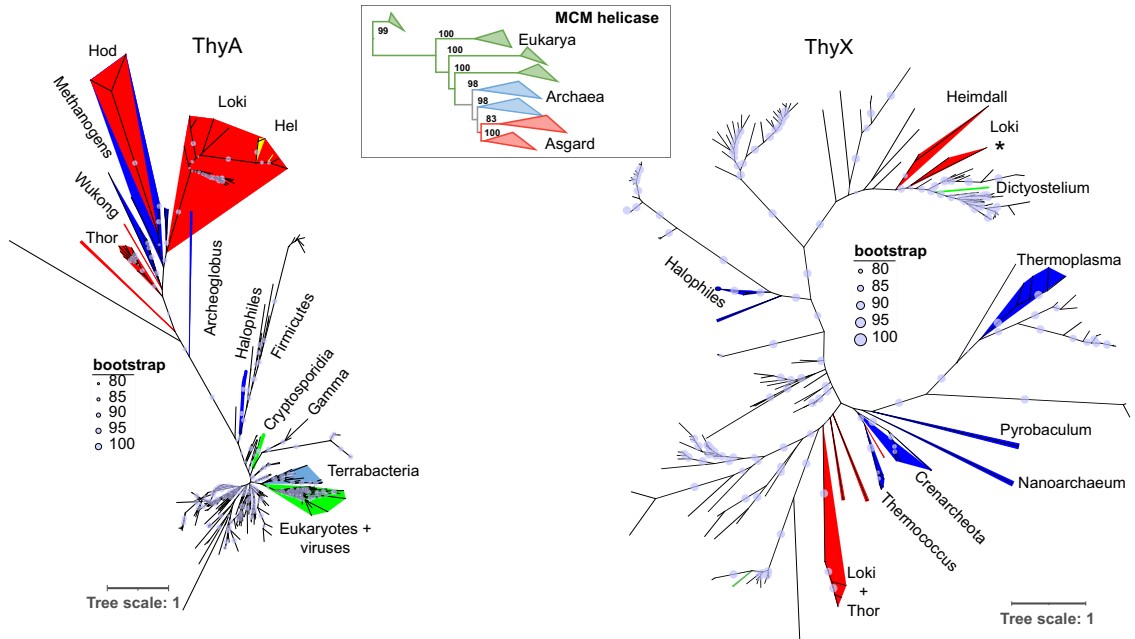

**Fig. 4 | Unrooted maximum likelihood phylogenetic tree of ThyX (198 sequences, alignment of 157 amino acids with the LG+G+I phylogenetic model) and ThyA (390 sequences, alignment of 121 amino acids with the LG+G +I model) with a focus on Asgard proteins.** The obtained ThyX tree indicates multiple lateral gene transfers from bacteria to *Psyn* and other Asgard genomes. By contrast, the ThyA tree refutes the occurrence of recent and multiple lateral gene transfers from bacteria to Asgard genomes. Asgard sequences are indicated in red, other archaea in blue, bacteria in black, and eukarya in green. The scale bar indicates the average number of substitutions per site. Bootstrap values are proportional to the size of the circles on the branches. The complete trees are available in the supplementary data. Insets represent the same trees for which nodes that have bootstrap supports lower than 85% have been collapsed. The MCM replicative helicase phylogeny was constructed as control as indicated in the text.

the Asgard ancestor. Notably, in ThyA phylogenetic trees, eukaryotic thymidylate synthases group consistently together with bacterial, and not archaeal sequences, which is supported by a bootstrap value of 100. (Fig. 4). ThyX and ThyA tree topologies were conserved and the conclusions remain valid even when the branches with less than 85% of the bootstrap support were deleted (Supplementary Fig. 6).

## ThyA and ThyX genes are inserted in archaeal-like genomic contexts

To exclude the possibility that *Asgard* metagenomes and genome assemblies are contaminated with foreign DNA sequences, we analysed the taxonomic origins of the genomic environments of the Asgard *thyX* and *thyA* genes against the NCBI reference sequence database (Fig. 5a). An important fraction of the genes surrounding *Asgard thyX* are unambiguously of archaeal origin (Fig. 5a): for one third of Asgard contigs, the majority of the best hits corresponds to archaeal sequences. A very similar pattern is observed for the Asgard contigs carrying *thyA* (Fig. 5a). By focusing on the genomic context of the *Psyn thyx* gene (Supplementary Fig. 1), we show that almost all of the 18 genes upstream or downstream have their first BLAST hit matching with an archaeon. Moreover, 14 of these genes have more than 75% of their top100 BLAST hits belonging to archaea. These observations indicate that most Asgard *thyX* and *thyA* genes present in (meta)genomes are inserted into archaeal-like genomic contexts and are unlikely to result from DNA contaminations or assembly artefacts.

To gain insight into the evolution of the genomic contexts of the *thyA* and *thyX* genes, we analysed the level of gene co-linearity (synteny) of the *thyA* and *thyX*-encoding contigs (Fig. 5b and c). Regarding *thyA* (Fig. 5b), three major groups emerge, displaying strong conservation of the gene order. In particular, Thorarchaeota, and the majority of Lokiarcheota and Hodarcheota form three distinct synteny clusters. The remaining *thyA* encoding contigs display low level synteny. By contrast, *thyX*-encoding contigs, with the exception of the Thor BC and Loki Rbin_111 contigs, display virtually no genome synteny (Fig. 5c).

## *Psyn thyX* is functional in *Escherichia coli*

Next, we investigated whether *Psyn thyX* functionally complements growth defects of an *E. coli* strain specifically impaired in thymidylate synthase activity. Towards this goal we designed and constructed a synthetic plasmid, pTwist-Psyn-ThyX, where the transcription of *Psyn thyX* is under control of a synthetic T5 promoter, for details on the inducible expression vector see Fig. 6a. As the T5 promoter is recognized by native *E. coli* RNA polymerase, this plasmid expresses an N-terminal His-tagged *Psyn* ThyX protein in any *E. coli* strain.

The ability of *Psyn thyX* to permit thymidine-independent growth of the *E. coli* thymidine-auxotroph strain FE013[11] (Δ*thyA::aphA3*, derived from wild type MG1655) was checked after three days at room temperature or 37°C in the presence of 1 mM IPTG (isopropyl β-D-1-thiogalopyranoside) using either minimal M9 or thymidine-deprived rich medium L⁺ (see the methods section). Figure 6b shows the formation of individual colonies of *Psyn thyX*/FE013 in the absence of thymidine in the presence of IPTG and appropriate antibiotics (plate on left, streaks 1 and 2). Under these thymidine limiting conditions, the strain carrying the control plasmid lacking the insert did not form individual colonies (streak 3, Fig. 6b). Altogether the results of this inter-kingdom complementation experiment indicate that *Psyn* ThyX protein is fully functional in a bacterial strain and must use *E. coli* folate derivatives for *de novo* thymidylate synthesis.

## Enzymatic activity of *Psyn*ThyX

Recombinant histidine-tagged *Psyn*ThyX was produced in soluble form and efficiently purified by one-step affinity chromatography (Fig. 7a). Western blot analysis using an anti-histidine-tag monoclonal antibody revealed a single band with an apparent molecular mass of ~37 kDa (Fig. 7b). ThyX is a dUMP-dependent NADPH oxidase[32,33], differently from ThyA. The formation of dTMP catalyzed by ThyX enzymes involves two half-reactions, oxidation and transfer of the methylene group from $CH_2H_4Folate$ to dUMP, to form the final

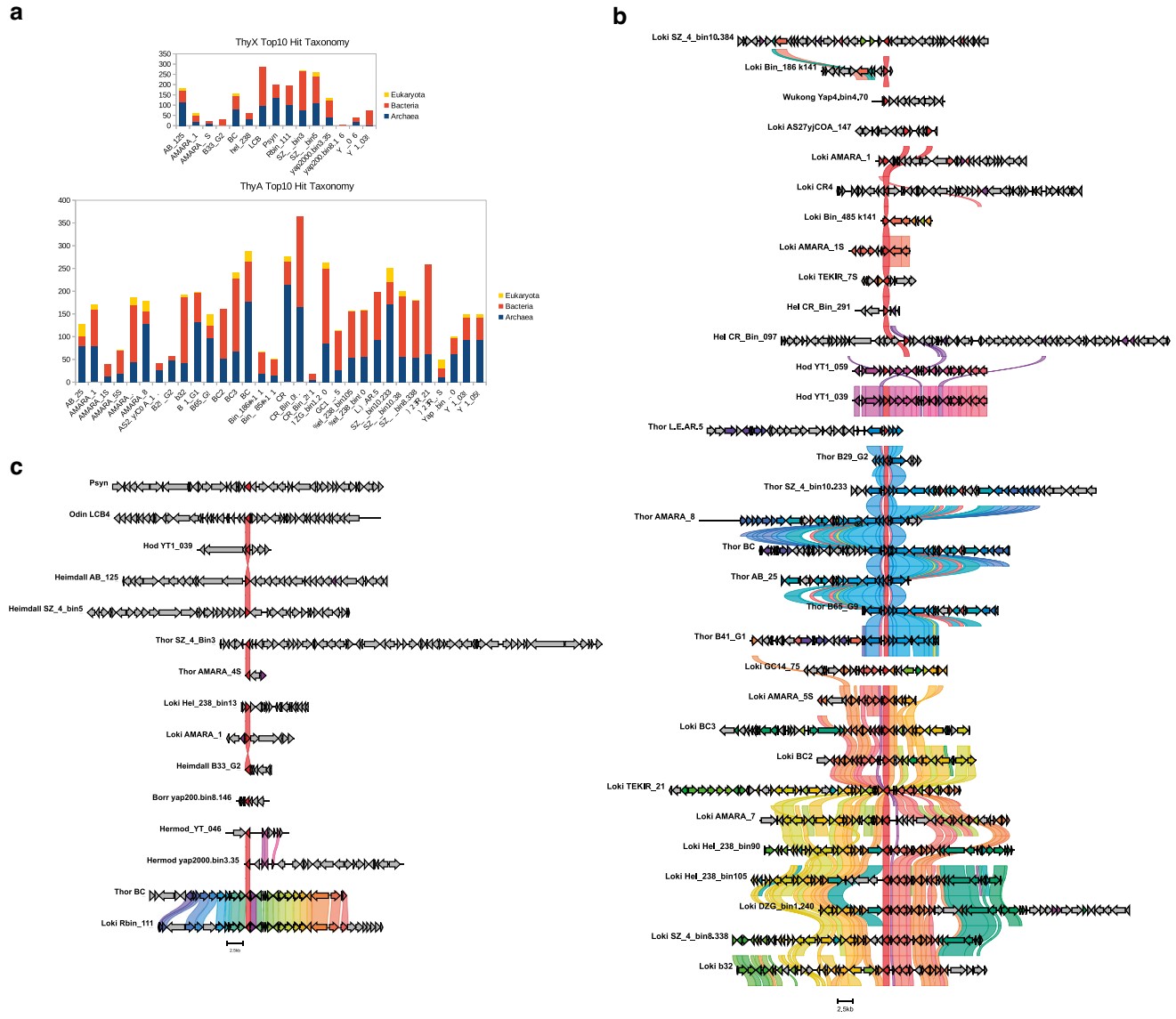

**Fig. 5 | Analysis of the genomic environments of Asgard *thyX* and *thyA* genes.** **a** Taxonomic origin of the neighboring genes of each Asgard contigs encoding *thyx* or *thyA* genes. The bar plots indicate the taxonomic origins of the best ten hits against the NCBI RefSeq database of each gene present in the respective contigs. Genes matching with Eukaryotic sequences have been indicated in yellow, in red for Bacteria and in blue for Archaea. **b, c** Synteny maps of the Asgard contigs encoding *thyA* and *thyX*. The colored tapes indicated homologous genes between the different contigs. Contigs have been sorted by their level of shared gene synteny and centered on the *thyx* and *thyA* genes (indicated in red).

product dTMP. Using a spectrophotometric biochemical assay, we found that *Psyn* ThyX catalyzes the NADPH oxidation to NADP⁺ only in the presence of dUMP (Fig. 7c). The oxidation of NADPH was assayed with 0.4 μM *Psyn*ThyX using saturating amounts of FAD cofactor (50 μM) and the substrates dUMP (20 μM) and NADPH (750 μM). The specific activity of *Psyn*ThyX (0.030 UI.mg-1) is somewhat lower than that of *Mycobacterium tuberculosis* ThyX (0.044 UI.mg-1) or *Paramecium bursaria chlorella virus-1* (PBCV-1) ThyX (0.043 UI.mg-1). However, this level of activity is sufficient for genetic complementation (Fig. 6b). Under these experimental conditions, the hyperbolic saturation curves show a nanomolar affinity of the nucleotide substrate for the enzyme [$K_{m, dUMP} = 235 \pm 35$ nM (Fig. 7d)].

### Inhibition of *Psyn*ThyX by folate analogs

Our genetic complementation tests indicate that *Psyn* ThyX must use bacterial folates for thymidylate synthesis. To provide additional experimental support for this notion, we further characterized

*Psyn*ThyX enzymatic activity by performing kinetics measurements in the presence of molecules that bind to the folate binding pocket of bacterial ThyX proteins. These inhibitory studies of *Psyn* ThyX were performed using $H_4$folate (reaction product), $CH_2H_4$folate (substrate), and the tight-binding *Mycobacterium tuberculosis* ThyX inhibitor 2716[34,35], which all inhibit the NADPH oxidase activity of bacterial ThyX. This analysis revealed that the three bacterial folate analogs tested substantially inhibit *Psyn* ThyX activity compared to an assay without the addition of any molecule (Fig. 7e). Note that molecule 2716, a potent inhibitor of *Mycobacterium tuberculosis* ThyX, was solubilized in DMSO and results need to be compared to the control condition in presence of 1% DMSO. The three folate analogs presented a percentage of inhibition of *Psyn*ThyX over 50% with high reproducibility (Fig. 7f). Altogether our genetic (Fig. 6) and biochemical (Fig. 7) data indicate that bacterial folate-like molecules are efficiently utilized by archaeal *Psyn* ThyX for *de novo* synthesis.

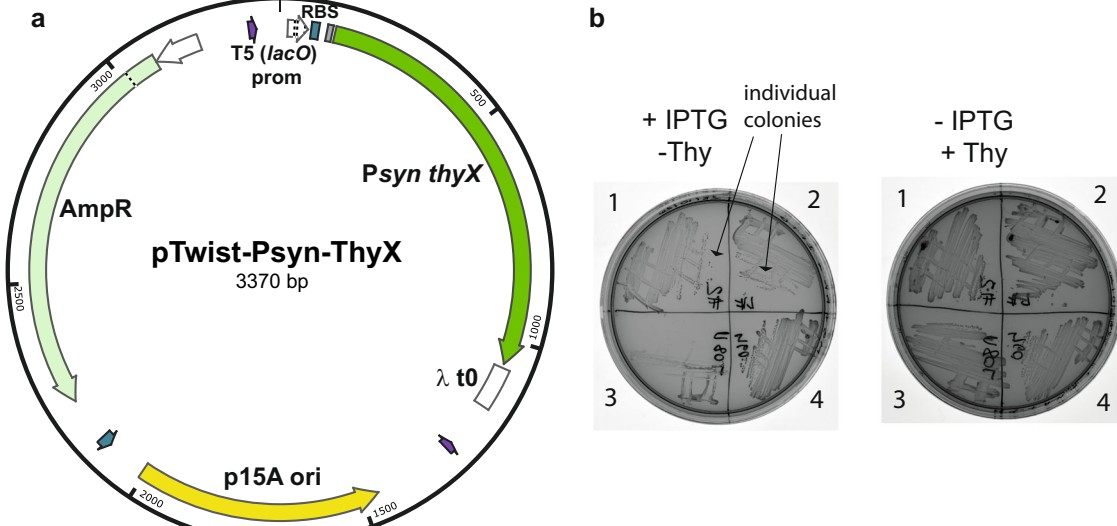

**Fig. 6 | Genetic complementation experiments using *Psyn* thyX. a** Map for the synthetic plasmid pTwist-Psyn-ThyX. Transcription of the *Psyn thyX* gene is under the control of a synthetic T5 promoter carrying *lacO*. An ampicillin resistance marker and a p15A replication origin are integrated, as well as a lambda *t0* terminator. **b** Interdomain complementation assay using *Psyn thyX*. The ability of *Psyn thyX* to permit thymidine-independent growth of *E. coli* thymidine-auxotroph

FE013[11] (Δ*thyA*, *lacI*) was checked after three days at 37 °C in the presence of 1 mM IPTG using minimal M9 medium. 1 and 2 correspond to two independent clones of *Psyn thyX*/FE013 whereas 3 and 4 are negative (no insert) and positive (*C. trachomatics thyX*) controls, respectively. The plate on the left contains IPTG, but no thymidine, whereas the plate on the right contains thymidine, but no IPTG.

## Distribution of the folate-mediated one-carbon metabolism enzymes in *Asgard* archaea

As folates are important biological cofactors that play a wide role in the biosynthesis of DNA, RNA, proteins and vitamins, our observations obtained using *Psyn T*hyX prompted us to investigate in more detail folate-dependent biosynthetic networks in genomes and metagenomes of Asgard archaea.

Our analyses indicated the sporadic presence of many $C_1$ generating/transferring and folate-interconverting enzymes in Asgard archaea (Fig. 2). In addition to *thyX* and *thyA* genes, we analysed the distribution of *folA*, 5-formyltetrahydrofolate cyclo-ligase (MTHFS), *folD*, methylenetetrahydrofolate reductase (MTHFR), *metH*, formate-tetrahydrofolate ligase (FTHFS), and *purH* genes in the *Psyn* genome and other Asgard metagenomic assemblies (Fig. 2 and supplementary Data 1). Many of these were predicted as likely transfer events by HGTector (Fig. 3 and Supplementary Fig. 3). ThyA appears the most universal folate-dependent *Asgard* enzyme, whereas the majority of the other folate-related genes have a very sporadic distribution. Nevertheless, the *folD* and *purH* genes are frequently found in *Asgard* archaea except for the *Heimdall* and *Odinarchaeota* groups, where they are absent or partial (sequence alignment length is less than 90% of the *Psyn* sequence).

The statistically robust phylogenetic analyses of these Asgard folate-dependent genes (Fig. 8 and Supplementary Figs. 7–13 for a detailed version of the trees with the taxon names) indicated that Asgard sequences appear as polyphyletic groups that often are scattered into bacterial subtrees, suggesting that the Asgard folate-related enzymes have participated in multiple gene transfer events from Bacteria (Fig. 8). The different Asgard sequences can also cluster together with other archaeal sequences, which is consistent with gene transfer between the different archaeal species.

## Reconstruction of the complete folate-mediated one-carbon metabolism network in *Psyn*

*Psyn* is unique among *Asgard* archaea, as this cultivated species contains a remarkable number of predicted folate-dependent enzymes (Fig. 2 and supplementary Data 1). This prompted us to perform a detailed reconstruction of enzymatic reactions participating in an

interdependent reaction network required for chemically activating and transferring $C_1$ units in *Psyn*. Automated genome-wide analyses, together with highly sensitive manual similarity searches, complement previous analyses on this topic[23]. *Psyn* has a high metabolic potential of to use folate derivates for de *novo* synthesis of not only thymidylate, but also of inosine-5'-monophosphate (IMP) and remethylation of homocysteine to methionine (Fig. 9a). Moreover, both carbon dioxide (at the level of 10-formyl THF) and serine (at the level of $CH_2H_4THF$) could provide a feasible source for one-carbon units entering the folate-mediated metabolic network in this species. The complete glycine cleavage system and its folate-binding component GcvT also provide an alternative source of $CH_2H_4$folate from glycine. According to these metabolic reconstruction studies, the folate network moreover participates in histidine catabolism by converting histidine to glutamate and 5-formyl THF. Many of these predictions are well supported by previously unreported associations of genes encoding folate-dependent enzymes that are often physically and/or functionally linked (Fig. 9b).

## Discussion

As detailed in the manuscript, our bioinformatics studies led to biochemical characterization of an archaeal thymidylate synthase. This characterization of an archaeal thymidylate synthase was important, considering that the sequence identity of *Psyn* ThyX with bacterial homologs varies between 40–50%. Indeed, at this level of sequence identity, transfer of functional annotations between homologs may result in incorrect inclusions in 5–10% of cases[36]. We also note that at least four different folate/FAD-dependent strategies for methylating or demethylating a wide range of substrates including DNA precursors, RNA molecules, eukaryotic histones, or antibiotics molecules do exist in biological systems[12], further underlining the importance of the reported experimental validation of substrate-specificity. Our genetic and biochemical experiments reveal that this enzyme functions as a robust thymidylate synthase in bacterial cells (Fig. 6). We also show that *Psyn* ThyX catalyzes dUMP-dependent NADPH oxidation, which is significantly inhibited by bacterial folate analogs (Fig. 7), as has been previously shown for bacterial or viral ThyX proteins (for a recent review, see[12]). Note that our inhibitors of archaeal ThyX could be

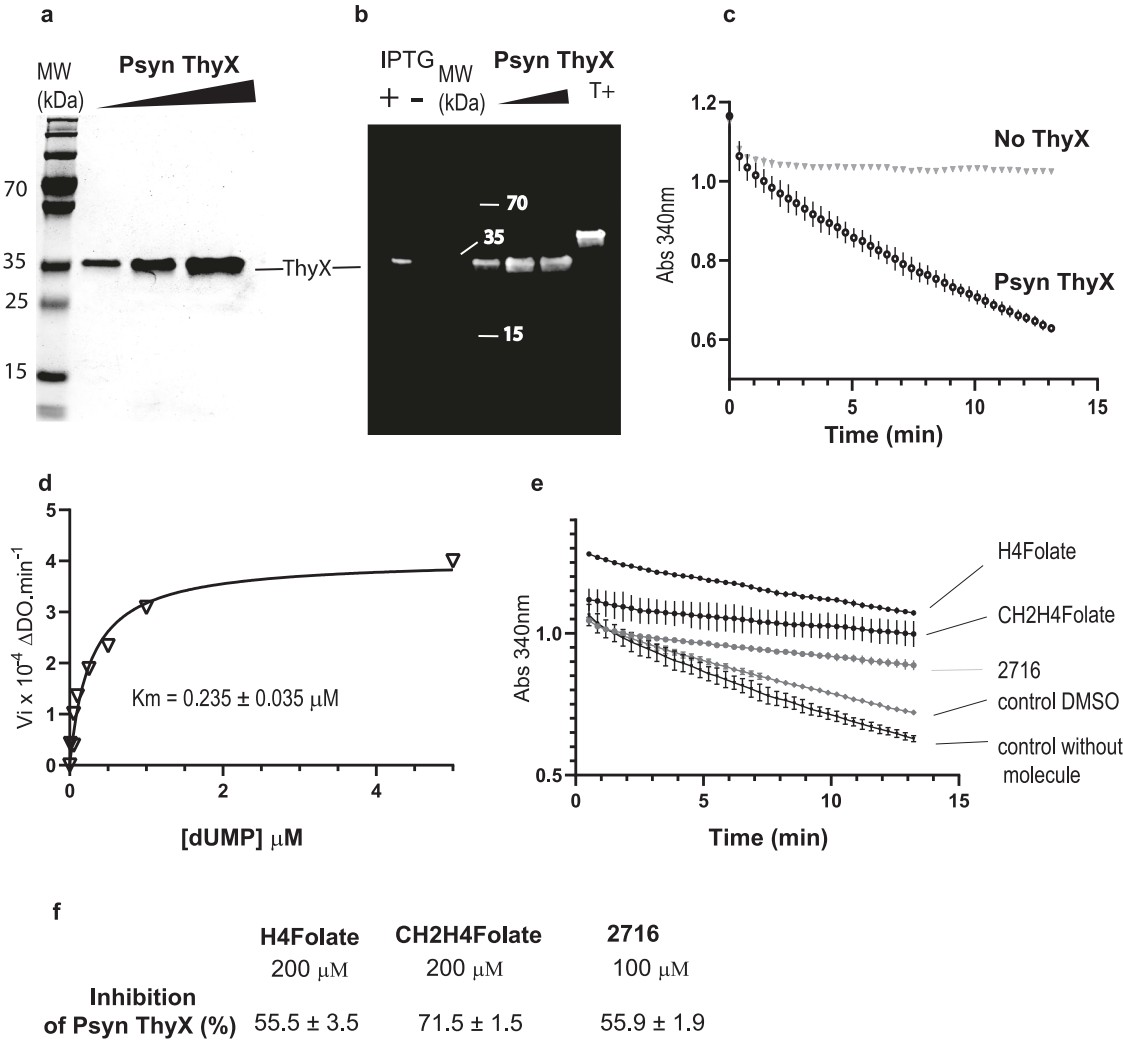

**Fig. 7 | Biochemical characterization and inhibition analyses of *Psyn* ThyX.**
**a** 12% TGX™ PAGE of *Psyn* ThyX purified by one-step affinity chromatography. 0.5, 1, and 2 micrograms of purified protein were loaded from the left to the right. MW: molecular marker. **b** Western immunoblot of IPTG-induced and non-induced whole cell lysates of *E. coli* FE013 using anti-His monoclonal antibodies. A single band with an apparent molecular mass of ~37 kDa is revealed. T+: control lane (*Pyrococcus abyssi* RNA ligase). Panels a and b show representative plots from two different biological replicates. **c** Representative NADPH oxidation activity curves of *Psyn* ThyX as measured following the absorption change at 340 nm. The control without

enzyme is also shown. Assay mixtures contained 0.4 µM enzyme, 50 µM FAD, 20 µM dUMP, and 750 µM NADPH. **d** Oxidation rate of NADPH under saturating dUMP concentrations. The affinity of dUMP for *Psyn* ThyX was determined from the hyperbolic saturation curve. **e** Evaluation of inhibiting properties of $H_4$folate (200 µM), $CH_2H_4$folate (200 µM), and the folate analog '2716' (100 µM)[23, 24]. Controls included the assays without the inhibitor in the presence of 1% DMSO. **f** Percentage of inhibition calculated for the three folate derivatives or inhibitors. In panels c, e and f data are presented as mean values ± Standard deviations (S.D) of three independent measurements.

valuable tools for determining the contribution of ThyA and ThyX containing subgroups of anaerobic methanotrophic (ANME) archaea to methane oxidation in mixed ANME communities[37]. Finally, considering that based on the experimental analyses the majority of archaea use chemically modified C1 carriers different from $H_4$folate derivatives[19,20], the reported biochemical observations suggesting the use of "bacterial" folates in Asgard is of interest.

Our phylogenetic studies suggest that Asgard archaeal *Psyn* ThyX originated in Bacteria (Fig. 4). An alternative explanation could be the possible contamination of the Asgard (meta)genome assemblies with foreign DNA. To exclude this possibility, we analysed the genomic environment of the Asgard *thyX* genes (Figs. 1 and 5, and supplementary Fig. 1). This analysis indicates that the Asgard *thyX* genes are inserted into an archaeal-like genomic context. Thus, these genes represent bona fide thymidylate synthase genes acquired laterally from bacteria. The distribution of *thyA* in the majority of Asgard metagenomes, and the phylogenetic clustering of thymidylate synthase genes (Fig. 4) along with other archaeal sequences and suggest

that *thyA* is the ancestral thymidylate synthase in the analyzed group of Asgard archaea. The high level of gene synteny of the corresponding genomic segments was detected for some Asgard *thyA* MAGs, but not for *thyX* sequences (Fig. 5b).

Taken together, our phyletic and phylogenetic analyses of thymidylate synthase genes in Asgard archaea reveal a complex evolutionary history involving many lateral gene transfers from bacteria and subsequent homologous and non-homologous replacements. This phenomenon is mediated in part by bacteriophages and/or archaeal viruses[13]. The non-homologous displacement of the *thyA* gene by *thyX* has already been suggested for diverse bacterial and archaeal species[2,13], but genetic (Fig. 6) and biochemical (Fig. 7) validation of the functionality of the acquired gene via HGT has been lacking. To further confirm these conclusions, we performed gene tree species tree reconciliation analyses using parsimony (Supplementary Figs. 14 and 15). In agreement with Fig. 4, for *thyX* these analyses indicated 2 bacteria-to-Asgard HGT, 3 intra-Asgard HGT, and 1 Asgard-to-Archea HGT. Regarding *thyA*, the reconciliation analysis predicted only

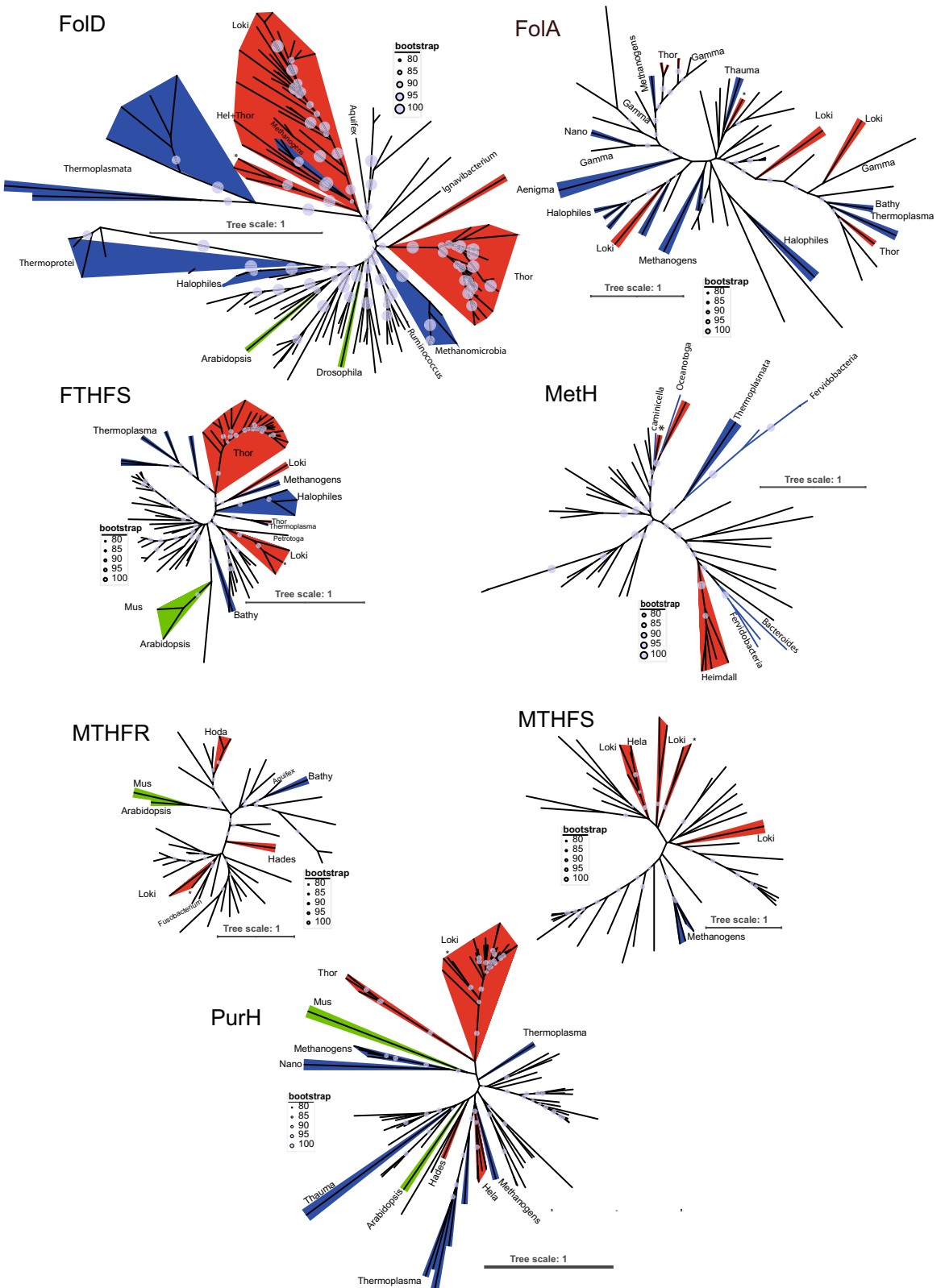

**Fig. 8 | Unrooted Maximum likelihood phylogenetic tree of the Asgard folate-related proteins.** FolA (83 sequences, alignment of 153 amino acids), FolD (130 sequences, alignment of 244 amino acids), MTHFS (56 sequences, alignment of 149 amino acids), FTHFS (79 sequences, alignment of 509 amino acids), PurH (94 sequences, alignment of 437 amino acids), MetH (47 sequences, alignment of 131 amino acids) and MTHFR (53 sequences, alignment of 280 amino acids) with a focus on Asgard proteins. The scale bar indicates the average number of substitutions per site. Bootstrap values are proportional to the size of the circles on the branches. LG+G+I model was used in all the other cases except for FolA (LG+G model). The complete trees are available in the supplementary data.

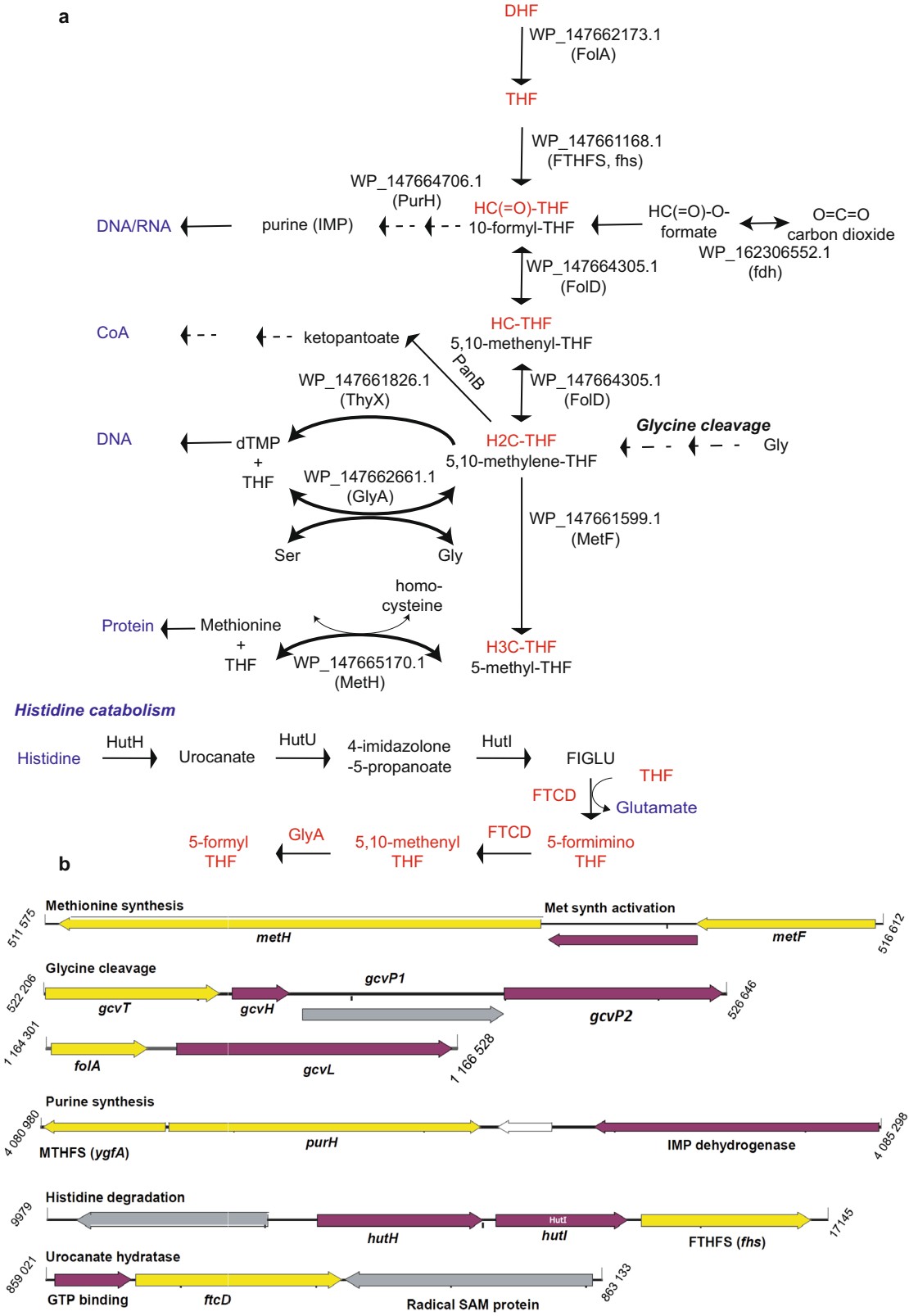

**Fig. 9 | Identification and genetic connections of folate-dependent enzymes in *Psyn*. a** Metabolic reconstruction of the complete *Psyn* folate-dependent metabolism. Abbreviations not explained in the figure are DHF dihydrofolate, THF tetrahydrofolate, IMP inosine monophosphate. The previously suggested pathways identified earlier (ref. [23]) are indicated in italics. **b** Functional connections between the folate-dependent *Psyn* enzymes (indicated in yellow) as suggested by the genomic context analyses. Purple and gray refer to genes functionally linked with folate-dependent enzymes. All these genes are moderately expressed in *Psyn* (Supplementary Fig. 2) For additional discussion, see text.

Asgard-to-Asgard HGT ($n = 5$) and Archea-to-Asgard HGT ($n = 5$). As these analyses used more Asgard *thyA* than *thyX* sequences, the detected numbers of HGTs transfers are not directly comparable. These data suggest that analyzed Asgard *thyA* sequences have either been predominantly inherited vertically or via HGT events within archaea. However, *Asgard thyX* genes have clearly participated in interdomain gene transfers (Fig. 4).

The phylogeny of *Psyn* ThyX, together with its biochemical properties, prompted us to investigate more globally the distribution and phylogeny of folate-dependent enzymes in Asgard. We detected many genes encoding for folate-dependent enzymes that are all transcribed in *Psyn* (Supplementary Fig. 2). This suggests that a complex interdependent reaction network operates to chemically activate and transfer one-carbon units allowing DNA, RNA, protein, and vitamin (pantothenic acid) synthesis in this species. In particular, a syntrophic amino acid utilization of *Psyn* using serine, glycine, and histidine is plausible, as these amino acids can readily provide one-carbon units for the folate metabolism at different oxidation levels (Fig. 9a). *Psyn* likely uses folate derivatives for the synthesis of thymidylate, purine, and methionine. In many cases, a strong association of the genes for folate-dependent enzymes with additional functionally linked genes, for instance, glycine cleavage or histidine catabolism, was also observed (Fig. 9b). In agreement with a previous study[23], biosynthetic pathways for folic acid and tetrahydromethanopterin appear incomplete in *Psyn*. However, the growth medium of *Psyn* is supplemented with folic acid-containing vitamin supplement[23], and many bacteria are known to transport folates[38–40]. These observations raised the possibility of *Psyn* cross-feeding with the folates produced by its symbiotic partners, as has been previously observed for bacterial communities[39,41].

Strikingly, Asgard folate-dependent enzymes appear to frequently participate in HGT events [Figs. 4 and 8 (*Psyn* genes are indicated by asterisks)]. The transfer of genes from Bacteria to Archaea is well documented[42]. However, our study has revealed a remarkable array of gene transfers that have potentially influenced many metabolic pathways of this enigmatic species. Our phylogenetic analyses did not suggest a well-defined single donor for these gene transfers that may have occurred on multiple and sequential occasions. By analogy with the observations made in bacteria[43], our observations agree well with the previous suggestion that HGT of metabolic networks may provide an adaptive benefit to *Psyn* in response to changing environments. Our observations are of general interest, as our HGTector analysis (Fig. 3) indicated that at least 4% of *Psyn* genes correspond to transfer events, including many genes implicated in amino acid and nucleotide metabolism. Automated HGTector cut offs may underestimate the true extent of HGT for this species, as the initial discovery of the first Asgard genome, a lokiarchaeal species, reported a substantial fraction of genes with bacterial origins [29% of all genes][44]. Widespread interdomain HGTs have also been proposed in uncultured planktonic thaumarchaeota and euryarchaeota[45]. Obligatory association of *Psyn* with co-cultured bacteria might have favored the gene exchanges, possibly influencing microbial population structure under natural conditions[46]. Several different archaeal molecular mechanisms facilitating cross-domain DNA transfer, including integrons[47],have been described[48], thus providing plausible means for HGT between Bacteria and Asgard. Finally, frequent HGTs between *Psyn* and different bacterial species might partially explain the apparent chimeric profiles of Asgard genomes[49], which seemingly result from the frequent mixing of ancestral archaeal genes with laterally inherited bacterial ones. It was recently suggested that some Asgard MAGs display evidence of assembly errors of ribosomal proteins in a concatenated alignments[49]. In the light of our analyses (Fig. 5), we think that our results do not reflect assembly errors of Asgard MAGs. Consequently, the frequent HGTs between *Psyn* and different bacterial species have likely contributed to evolutionary dynamics of Asgard genomes,and our study

on the thymidylate metabolism might represent only the tip of an iceberg of functional "bacterial" genes in Asgard genomes.

Our analyses indicate that eukaryotic ThyA does not directly originate from Asgard archaea (Fig. 4). In the context of the different eukaryogenesis models, this observation is compatible with two possible scenarios. First, Asgard archaea may have contributed their ancestral thymidylate synthase to the ancestor of eukaryotic cells, followed by its sequential replacement by a bacterial ThyA. Alternatively, bacterial symbionts may have directly provided their *thyA* to the ancestral eukaryotic cell. Both scenarios suggest that the thymidylate metabolism in Eukarya relying on "bacterial" C1 carriers was transferred from bacteria. Consequently, eukaryotic cells require both "archaeal replisome" and "bacterial metabolic enzymes" for replicating their genetic material.

Taken together, our data support the idea that Asgard archaea harbor highly mosaic genomes that have received many bacterial genes, even for crucial metabolic processes like the thymidylate and folate metabolism. We have also experimentally demonstrated that some of the transferred *Psyn* genes, like *Psyn thyX*, are fully functional in vivo and in vitro. More general studies on Asgard evolutionary dynamics are now needed to better appreciate the functional and evolutionary importance of the observed interdomain gene transfer between Asgard and bacteria, as well as the origin of the striking eukaryotic gene repertoire in this fascinating group of prokaryotes.

## Methods

### Bioinformatics methods and genome-wide analyses

The complete genome sequence available for the *Candidatus* Prometheoarchaeum syntrophicum strain MK-D1 (*Psyn*) chromosome (NZ_CP042905.1) was used for bioinformatics analyses. This genome is 4278 Mb with a GC content of 31.2%. RPKM values for gene expression were determined using the Geneious Prime® 2021.1.1 (Build 2021-03-12) and agreed well with the previously published values. Metabolic reconstructions were performed using the RAST server and KEGG databases. These analyses were complemented with highly sensitive manual similarity searches using HHpred[50].

### Structural alignments and structure modelling

PROfile Multiple Alignment with predicted Local Structures and 3D constraints (PROMALS3D) was used to align the *Psyn* ThyX sequence with the *Thermotoga (T.) maritima* ThyX structure (PDB 5CHP). PROMALS3D (http://prodata.swmed.edu/promals3d/promals3d.php) aligns multiple protein sequences and/or structures, with enhanced information from database searches, secondary structure prediction, and 3D structures. Input sequences and structures used FASTA and PDB formats, respectively. (red: alpha-helix, blue: beta-strand).

A model of the *Psyn* ThyX homotetramer was constructed with the protein structure modeling program Modeller using PDB structures 1O26, 3N0B, and 6J61 as templates[26]. Ten initial models were constructed and the best one was chosen using molpdf and DOPE score, and by evaluating the stereochemical quality with PROCHECK. A structural superposition of the model with PDB structure 4GT9 using the UCSF Chimera 1.15 software[27] allowed the addition of FAD, dUMP and folate molecules. To improve the structural model obtained, molecular dynamics simulations were performed with CHARMM[51] and Namd[52] molecular mechanics softwares.

### Automated prediction of HGT

Automated analyses of potential HGT events were performed using a HGTector[28]. This method first performs Diamond all-against-all similarity searches of each protein-coding sequence of *Psyn*. The NCBI genome database was downloaded and compiled (on August 2021) on a local machine. The output of the similarity searches was passed to the analysis program HGTector where the taxonomy of the self-group was defined as *Candidatus* Prometheoarchaeum

syntrophicum (NCBI TaxID: 2594042). The close-group was automatically inferred by HGTector as the "superkingdom Archaea" (TaxID: 2157) that contains 1062 taxa in the reference database. From the statistical distribution of close and distal hits, HGTector predicted a list of potentially transferred genes. The output of the program is a scatter plot, where the horizontal axis represents the close score, the vertical axis represents the distal score, and potential HGT genes are colored in yellow.

## Phylogenetic analyses

For phylogenetic analyses, we constructed a database with a backbone of 183 *thyX* sequences that cover all the major bacterial and archaeal phyla derived from a previous study[13]. Homologs found in Asgard genomes were then used as seeds for reciprocal BLAST searches against an NR database (15/01/2021 version) to find the best homolog outside Asgards. Finally, identical sequences (same identifying code or 100% sequence similarity) were removed to generate the final data set. We used similar approaches with the other folate enzymes to find homologous sequences in Asgard genomes. Phylogenetic analyses were performed using the obtained protein sequence data sets (see above) that were aligned using MAFFT v7.388 with default settings[53]. Ambiguously aligned sites were removed using trimAl v1.4.rev15 with the "-automated1" and "-phylip" options. The phylogenetic position of the Asgard sequences was then inferred with two different Maximum likelihood softwares: PhyML v. 1.8.1 (Supplementary Fig. 13 for ThyA and ThyX) and IQTREE[54] using the best substitution models as determined by Smart Model Selection[48] and ModelFinder[55], respectively. 1000 bootstrapped data sets were used to estimate the statistical confidences of the nodes using the Ultrafast option as implemented in IQTREE and the non-parametric bootstrap option in PhyML. Trees were then visualized using the iTol software[56].

To investigate which branches the transfer events most likely involved, reconciliation analyses were computed with ecceTERA[57]. SylvX[58] was used to visualize the reconciliation trees computed by ecceTERA. For reconciliation analyses, species trees are based on 16S rDNA phylogenies with the corresponding taxa in the ThyA and ThyX gene trees. This led to reduced sequence data sets of 119 taxa for ThyA (including 14 Asgard sequences) and 96 taxa for ThyX (with 6 Asgards). IQ-TREE was used to build reference and gene trees with the following best models as predicted by ModelFinder: TVMe+I+G4 (16S) and LG+I+G4 (ThyX and ThyA).

## Analysis of the genomic environments of the ThyA and ThyX genes

Metagenomic Asgard contigs encoding *thyA* and *thyX* genes larger than 10 kb were retrieved and gene annotation were carried out using RAST[59]. The taxonomic group of each protein-encoding genes was determined as follows : similarity searches against the NCBI RefSeq database (02/09/2021 version) were computed using Diamond[60] with blastp option and e-value cuttoff of 1e-10. The ten best hits of each gene were retrieved and the taxonomic profiling of these hits (Bacteria/Archaea/Eukarya) was collected using the PYlogeny software (https://github.com/jrjhealey/PYlogeny).

For analysis of the genomic environment of the *Psyn thyX* gene, a more in-depth analysis was carried out by collecting all the genes located 15 kb downstream and 15 kb upstream of the *Psyn thyX* and searched for homologs with BLASTP against an NR database (15/01/2021 version). The first BLASTP hit was retrieved, as well as the percentages of the archaeal sequences in the taxonomy profile of the first top 100 hits.

Analysis of the synteny of the Asgard contigs encoding *thyA* and *thyX* genes were conducted using the Clinker package[61] with default parameters except the minimal identity thresholds that have been reduced to 25% (« -i 0.25 » option).

## Design and synthesis of Psyn ThyX expression plasmid

pTwist-Psyn-ThyX plasmid (3370 bp), carrying the synthetic Psyn *thyX* gene of 897 bp was synthesized by Twist Biosciences (https://www.twistbioscience.com) and confirmed by sequencing. In the final design, the target gene *Psyn thyX* was placed under the control of a strong bacteriophage T5 promotor carrying *lacO* sites. This construct also carried an appropriate ribosome binding site, a lambda t0 transcriptional terminator, and the Amp$^R$ gene. The pTwist-Psyn-ThyX plasmid uses a p15A origin of replication and contains the codon-optimized synthetic gene of *Psyn*-ThyX with an N-terminal 6xHis sequence.

## Genetic complementation tests

The ability of *Psyn thyX* to permit thymidine-independent formation of individual colonies of the *E. coli* thymidine-auxotroph FE013 strain[11] (Δ*thyA::aphA3 lacI*, derived from wild type MG1655 strain) was checked after three days at room temperature or at 37 °C in the presence of 1 mM IPTG (isopropyl β-D-1-thiogalopyranoside) using either minimal M9 (shown in Fig. 6) or thymidine-deprived enriched medium L$^+$. Eight individual colonies all demonstrating similar phenotypes in the absence of thymidine were tested.

## Protein purification

*E. coli* FE013 cells carrying pTwist-Psyn-ThyX were grown at 37 °C and shaken at 150rpm in liquid Luria Bertani medium supplemented with ampicillin (final 100 μg/mL) until an optical density of 600 nm around 0.7 was reached. Expression of recombinant *Psyn* ThyX was induced by the addition of 0.5 mM IPTG during 3 h at 37 °C. Cells were harvested by centrifugation at 6000 x g at 4 °C for 30 min before storage at −20 °C. The *Psyn* ThyX recombinant protein was purified on Protino Ni-TED column (Macherey-Nagel) as previously described[62]. Stepwise protein elution was performed with imidazole at 50 mM, 100 mM, 150 mM and 250 mM in a phosphate buffer (50 mM Na$_2$HPO$_4$-NaH$_2$PO$_4$, pH 8, 300 mM NaCl). The 100 mM imidazole fractions containing the *Psyn* ThyX enzyme were pooled, buffer exchanged on Econo-Pac PD-10 Columns (Bio-Rad Laboratories, Hercules, CA) with phosphate buffer (50 mM Na$_2$HPO$_4$-NaH$_2$PO$_4$, pH8, 300 mM NaCl), concentrated to a final concentration of 4 μM and stored at −20 °C in the same phosphate buffer complemented with glycerol (10% v/v).

The purified proteins were analyzed on 12% TGX Precast Gels (Bio-Rad Laboratories, Hercules, CA) followed by Coomassie Brillant Blue staining. For immunoblotting detection, after protein transfer, the nitrocellulose membranes (Bio-Rad) were treated with 20% blocking buffer (Li-Cor Biosciences, Lincoln, US) followed by incubation with anti-His-tag mouse primary antibody (dilution1/5000, Bio-Rad) and revelation with an IRDye 800CW Goat anti-mouse IgG secondary antibody (dilution 1/10000, Li-Cor Biosciences) using a ChemiDoc™ Touch Imaging System (Bio-Rad).

## Enzymatic activity tests

The NADPH oxidase assay consists of measuring the conversion of NADPH to NADP$^+$ via the decrease in absorbance at 340 nm. During the test, in 96-well plates, 100 μL of the standard reaction mixture (50 mM buffer Na$_2$HPO$_4$-NaH$_2$PO$_4$ pH 8, NaCl 300 mM, MgCl$_2$ 2 mM, FAD 50 μM, β-mercaptoethanol 1.43 mM, dUMP 20 μM, NADPH 750 μM, glycerol 8%, 0.4 μM of purified *Psyn*ThyX) were incubated in a Chameleon II microplate reader (Hidex) at 25 °C. Measurements took place over 15 min and were done in triplicates. The enzyme-free reaction was used as a negative control.

Tests of ThyX inhibition were performed by incubation of molecules at 100 or 200 μM with *Psyn*ThyX enzyme (0.4 μM) in the standard reaction mixture, without NADPH, for 10 min at 25 °C before starting measurements by automatically injecting NADPH. The molecule 2716 was solubilized in dimethylsulfoxide (DMSO) and used at 1% final concentration of DMSO during the test. % of inhibition was calculated using the following equation: $((Vo-Vi)/Vo)*100$; Vo and Vi are,

respectively, the initial rates of the reaction without or with addition of the molecule to the assay.

## Reporting summary

Further information on research design is available in the Nature Portfolio Reporting Summary linked to this article.

## Data availability

The bioinformatic data generated in this study have been deposited in Figshare   https://doi.org/10.6084/m9.figshare.20067425.v2).   These include results from sequence alignments, Blast searches and reconciliation analyses, IQTree and phyML trees as well as contig annotations, taxonomy data and protein accession numbers. The biochemical data for Fig. 7 are provided in the Source Data file. Source data are provided with this paper.

## Code availability

We slightly modified the source code of the HGTector which was made publicly available at https://github.com/cgneo/neoHGT. These modifications did not influence HGT predictions but fixed some minor issues that increased the stability of server connections.

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

## Acknowledgements

We thank P. Forterre for discussions on the evolution of DNA as a genetic material and Y.I. Wolf (National Center for Biotechnology Information, U.S. National Library of Medicine, National Institutes of Health) for advice and guidance in analyzing Asgard (meta)genomes. Z.L. is a student of the Bachelor of Science program of E. Polytechnique. H.M., H.B. and U.L. thank CNRS, INSERM, and E. Polytechnique for their financial support. RZ has received funding from the European Union's Horizon 2020 research and innovation programme under the Marie Skłodowska-Curie grant agreement No 899987. W.Y. is a recipient of a doctoral grant from China Scholarship Council (CSC).

## Author contributions

J.F., H.B., U.L., and H.M. conceived the study and wrote the manuscript with contributions from all other authors. J.F., R.Z., W.Y. and H.M. performed phylogenetic and bioinformatics analyses and visualized phylogenetic trees. H.B. and L.M. provided biochemical data, and U.L. and H.M. performed genetic experiments. J.C.L. provided a structural model and Z.L. performed HGTector analyses. All authors read and commented on the manuscript.

## Competing interests

The authors declare no competing interests.
