## [Peer Review File · Nature Communications]

Bacterial Origins of Thymidylate Metabolism in Asgard Archaea and EukaryaReviewer #1 (Remarks to the Author):

In this manuscript, Filée and collaborators study the evolutionary history of folate metabolism in Asgard archaea. They particularly focus on the enzymes for thymidylate biosynthesis, for which they perform phylogenetic and genomic analyses, as well as functional characterization through expression in *E. coli* and biochemical analysis. The authors also expand their phylogenetic work on other enzymes that use folate.

This study contains a promising series of details and analyses. However, the general picture it recovers does not seem to provide a firm basis for the general themes that the authors outline, namely novel understanding into eukaryogenesis, or broad insights on the ecology and evolution of Asgard archaea. The bioinformatic analyses are not conclusive and informative enough to accompany the author's conclusions. Finally, although the manuscript is generally well written, it contains many errors, unclear sentences and dubious arguments. I do, however, believe that the authors have made interesting progress on Asgard archaeal molecular biology, and I fully encourage them to keep building onto this topic.

Here I include some of the major issues I encountered in this manuscript:

1. Relevance and support to conclusions

The authors contextualize their study by touching some of the current major questions around Asgard archaeal biology. Indeed, Asgard archaea are fascinating organisms that are bringing much anew on our ideas of eukaryogenesis, and are thus attractive objects of study. However, this study does not seem to add on eukaryogenetic models, nor it adequately provides general insights onto Asgard archaeal ecology and evolution.

Eukaryogenesis. The authors do explicitly want to include evolutionary models of the evolution of thymidylate synthesis between the Asgard archaea and Eukaryotic groups, but very little progress is made on this front, other than suggesting a non-Asgard archaeal origin of ThyA and ThyX in eukaryotes. This question could be tackled much better. Moreover, I believe the authors should more explicitly explain why this is a specifically interesting question in the first place.

General insights onto Asgard archaeal ecology and evolution. The authors include several general conclusions around this theme: (1) Asgard archaea contain plastic genomes with large impact of horizontal transfer, (2) ThyA is ancestral while ThyX has been recently incorporated multiple times through horizontal transfer, (3) folate metabolism in Prometheoarchaeum includes a variety of folate-dependent enzymes that have been obtained in Asgard archaea from bacteria and may be key to syntrophic interactions. Unfortunately, none of these points seem to be strongly supported (see also below).

2. Results: phylogenetic analyses

I have struggled to find a good way to provide feedback for several of the results. Particularly, the phylogenetic analyses are very difficult to interpret. They form the backbone of the present study, and yet the information provided is highly insufficient for adequate evaluation.

First, the followed Methodology does not provide all necessary details. Multiple key details are unclear, including how the authors downsample the initial dataset (phylum or genus?), how exactly the authors identify homologs in Asgard archaeal genomes, what the purpose of the reciprocal Blast search outside of Asgard archaeal sequences is, how the authors proceed once the final dataset is obtained (i.e. were the sequences aligned to the previous alignment, or re-aligned and trimmed de novo?). The authors then claim to have reconstructed maximum-likelihood trees using two different software, but they only show one tree per gene. The authors also claim to have performed "1000 bootstrapped data sets", but it is unclear under exactly which conditions, as PhyML and IQ-Tree provide different approaches to this. More importantly, the authors need to clarify, for trees obtained with IQ-Tree, whether these replicates correspond to the "Ultrafast bootstrap" or the "Non-parametric bootstrap" algorithms. As an additional note, the use of "Smart

model selection" as implemented in PhyML is also not state-of-the-art, as the models that are implemented in this method cannot deal with more complex, accurate models as the ones provided in IQ-Tree (e.g. Q.pfam, free-rate heterogeneity parameters, mixture models).

Importantly, the trees are displayed in unclear formats. The authors provide rooted trees for most of their phylogenies: how were these roots selected? Additionally, providing support values as circle sizes is very poor practice, as this really does not allow proper distinction of the corresponding values, which dramatically hampers adequate interpretation of all supplementary phylogeny figures. This is particularly important in the possible case that the authors have used "Ultrafast bootstrap" values in IQ-Tree. It would also be helpful that the authors provide an adequate color legend for all tree figures, and that this is complete (what are the different shades of blue? What does yellow represent?)

Regarding interpretation (on top of the previous issues)), it would be helpful that the authors clarify why *Dictyostellium* and *Alvinella* are the only eukaryote sequence found in the ThyX tree.

3. Results: horizontal gene transfer detection

The authors provide potentially interesting insights onto the horizontal gene transfer patterns of Prometheoarchaeum, which could help inform evolutionary models on evolutionary dynamics of Asgard archaea. However, the results provided from HGTector are not conclusive enough for sweeping statements. First, more information should be provided about the statistical robustness of the obtained candidate genes and, in its absence, deeper confirmatory analyses are needed (phylogenetic or synteny analyses are definitely good places to start). The authors also present a modified version of this software, but do not explain what those changes affect, thus requiring a reader to refer to Github to interpret results. Very importantly, the provided results can be severely compromised if we consider that the "Distal" comparisons may include eukaryotic sequences – which may be closer relatives to Asgard archaeal sequences than most archaeal sequences.

4. Results: synteny analyses and ancestry claims

These are a good complement to horizontal gene transfer detection. Moreover, these analyses are used by the authors as support to establish evolutionary dynamics of ThyA and ThyX. The difference in synteny conservation between these two sets of homologs is apparent, however insufficient data is provided for this interpretation. First of all, these results are rarely placed together with phylogenetic results. For example, the authors claim rampant horizontal transfer of ThyX as derived from their lack of synteny, but what is the explanation for the lack of synteny between gene copies that indeed appear as closest relatives in the ThyX phylogeny in Fig. 4? (This is also made difficult by the apparent lack of congruency between the genomes displayed in Fig. 5C and Fig. 4: where are the Hod, Borr Hermod and Odin sequences in Fig 4?).

Importantly, the synteny results for ThyA do show a certain degree of gene neighbourhood conservation, but this is purely restricted to groups of closely related genomes (e.g. no conserved synteny is detected between Thor and Hod, or between Thor and Loki). This may indicate vertical inheritance within these groups (are these genomes even representative of the entire groups?), but there is no indication of vertical inheritance prior to the divergence of these groups. Thus, the author's conclusion that ThyA was present in the ancestor of Asgard archaea is unsupported by these results or by the phylogenetic results in Fig. 4.

Non-exhaustive minor comments:

- Suppl. Fig 1: it would be useful to know how many of the top hits correspond to closely related genomes (i.e. true verticality signal).
- L. 306-308. Also not found in Hermod, Baldr, Hod, Kari and Gerd. It would be interesting to further discuss the relevance of the patchiness of these genes in Asgard archaea. What does this

mean for their biosynthetic capabilities, syntrophy potential and in general the evolution of their metabolism? Very importantly, what is the meaning of the substantially larger representation of these genes in Prometheoarchaeum, even in comparison to its closest relatives? Do these results indicate that the folate-dependent metabolic networks in Asgard archaea are generally quite limited, given the predominance of gene absences rather than the opposite?

Reviewer #2 (Remarks to the Author):

In this paper titled "Bacterial Origins of Thymidylate and Folate Metabolism in Asgard Archaea" the authors Filée et al., have performed a detailed and comprehensive analysis of the folate metabolism in Asgard archaea. This newly discovered group has an interesting placement within the context of eukaryogenesis and hence has garnered much interest although the present analysis does not pertain to this but rather microbial biochemistry which is refreshing.

I only have some minor suggestions which the authors can consider.

Did the authors consider performing Gene-tree to species tree reconciliations maybe using ALE in order to further dissect the source and history of transfers in the ThyX and other genes? This could help resolve some uncertainty in the conclusions that have resulted from HGTector. It is potentially easy to check and only needs a species tree which should be easily constructable.

The authors mention the study by Garg et al, to say that they (Garg et al.) concluded that Asgard MAGs display evidence of assemble errors but as far as I understand they only looked at ribosomal proteins which may cause an issue when used in a concatenated alignment. This would be important for the authors when they do indeed do the reconciliation analyses. However, these should be mitigated by the inclusion of high-quality genomes. I would advise the authors to not misrepresent studies and inflate or deflate claims in a bad stroke.

The phylogenetic trees appear to be of a low quality and a little difficult to read even in the excellent ppt of the figures provided. Maybe these should be checked or re-colored before publication to be color blind friendly. Also rectangular representations tend to be more accurate and easier than circular notations.

In summary, I would like to see the results of a reconciliation analyses which would enrich what is already an extremely competent study. Thank you

Reviewer #3 (Remarks to the Author):

See the attached pdf for my comments and suggestions. I recommend publication after minor revisions

Reviewer #3 Attachment on the following page.

The article entitled “Bacterial Origins of Thymidylate and Folate Metabolism in Asgard Archaea” by Filée et al describes an extensive bioinformatic search for an archaeal flavin dependent thymidylate synthase (FDTS) and both the identification, purification and partial biochemical characterization of the enzyme. This discovery is significant and represents the first demonstration of an archaeal FDTS which has important evolutionary implications. The recommendation of the reviewer is to **publish after minor revisions** that are noted below.

Before providing my suggested improvements, I must disclose that I am a mechanistic enzymologists whose training is in the fields of protein biophysics/biochemistry and bioorganic chemistry. So I was unable to provide a critical review of the bioinformatic and phylogenetic analysis other than to say the authors should be commended for describing these parts in a manner that a non-expert can easily follow. I also am comfortable recommending publication because the authors’ analysis clearly found an archaeal FDTS. The enzymology was sound and the authors did *not* overstate the kinetic data as I often see without the proper controls.

Substernal but still Minor Changes Needed

1. **Figure 1 and throughout the text:** *Enzymes* carry out each of the reactions mentioned not the genes that encode for them. ThyA, FolaA and GlyA should be changed to thymidylate synthase (TS), dihydrofolate reductase (DHFR) and serine hydroxymethyl transferase (SHMT), respectively. Listing the name of the enzymes is more customary for both biochemists and biophysicists. It was distracting to constantly read the names of the genes especially for the thymidylate synthases. Call the classical one TS or TSase and the flavin dependent one FDTS or FDTase (the reviewer suggests leaving off “ase” for simplicity).

This is significant to improve the scientific rigor of the article but an easy edit to make. The *thyX* gene for example is expressed regardless of if FAD is present or not. Also, ThyX-FAD is written in Scheme 1 over the reaction arrows. This gives the impression that the flavin acts as a co-substrate and that binds, reacts and is released in a manner similar to the nicotinamide. This is not the case; FAD is non-covalently bound but remains associated to the enzyme throughout the catalytic cycle as is common in flavoenzymes.

The text will need to be fixed throughout the manuscript to more clearly denote the enzymes when they are the molecules being discussed and to not confuse them with the genes.

2. **Line 36:** Two additional references should be included for FDTS discovery and mechanism as listed:

a. “Mishanina, T.V., Yu, L., Karunaratne, K., Mondal, D., Corcoran, J.M., Choi, M.A., Kohen, A., *Science*, **2016**, 351, 507-510.”

b. “Koehn, E.M., Fleischmann, T., Conrad, J.A., Palfey, B.A., Lesley, S.A., Mathews, I.I., and Kohen, A.*, *Nature* **2009**, 458, 919-9233

3. **Lines 39-41:** This requires editing since it is confusing as written. “...accommodates dUMP, NADPH, and the carbon donor methylene tetrahydrofolate (CH₂H₄folate, a vitamin B9 derivative), is located at the interface of three...”

Change to: “...accommodates dUMP, NADPH, and the carbon donor methylene tetrahydrofolate (CH₂H₄folate, a vitamin B9 derivative). CH₂H₄folate is located at the interface of three subunits... It has been six or seven years since I have studied the structure of FDTS so am unsure if the authors intend to say that CH₂H₄folate is at the interface or that all three substrates (CH₂H₄folate, dUMP and folate) are at the interface. Please clarify.

4. **Line 55:** “The ThyX reaction mechanism **appears** less catalytically efficien...” FDTS is known to be less efficient since it suffers from significant substrate inhibition. While the k_{cat}/K_m values are roughly similar the significant substrate inhibition makes FDTS a more sluggish enzyme. Since it would distract from the impressive work here to cite the kinetic papers the authors should simply remove appears and state: “The **FDTS** reaction mechanism **is** less catalytically efficient...”

5. **Line 74:** Perhaps this is because I am not a molecular biologist but this completely unclear to me. What exactly was done in reference 3? I assume (did not check the ref at all) they knocked out the gene and visualized on a gel? Functional *thyX* to me means that FDTS was expressed and was able to make dTMP. A few words on what is meant by genetic evidence would help to clarify.

7. I think but am not certain *Prometheoarchaeum syntrophicum* should be in italics. Please check and correct if needed.

8. **Line 134:** This is not my area of expertise but the casing must surely be wrong here. If so please correct. If not please excuse my ignorance: “Reads Per Kilobase of transcript, per Million mapped reads”.

9. **Line 189 entire section:** I must admit as a mechanistic enzymologists this is entirely out of my field. I read through all of the genomic data and was able to clearly see what the authors did and were presenting but have no expertise to judge the merit of the work.

10. ***Psyn thyX* is functional in *Escherichia coli*:** The description of the plasmid is too long especially considering the plasmid map is shown in Figure 6a. The authors simply need to note it has a *lac* promoter, the gene to express the his tagged protein and that more details are shown in Figure 6a. This does not require 8 lines of text.

Also the termed “scored” after 3 days suggests a quantitative measure of cell growth was taken. This was not done and is not needed to make the authors point. “Scored” should therefore be changed to “checked”. This should also be changed on **line 507** in the methods.

Finally, please note in the text that both plates shown were grown 37 °C. I wondered if the additional growth observed on the plate to the right was due to temperature or the addition of thy. It was listed in the Figure legend but I did not see it in the text and this is important to stress the point the authors demonstrated.

11. The material and methods needs substernal proofreading. I am only going to list a few example here. The most obvious is that a space must be indeted after the number and before the unit. For example 50 mM and not 50mM. I noticed this in the results too and it should be fixed. It became annoying in the emethods because the authors were not consistent and sometimes even spelled out the number and the unit.

Line 481: typo. environments is misspelled.

Line 508: IPTG needs to be spelled out as isopropyl β-D-1-thiogalopyrnoside the first time it is mentioned. Delete (shown in Fig 5) this belongs in the results not the methods.

Line 513: light is not being absorbed but instead one is monitoring growth by looking at the turbidity of the media. Therefore, it is the *optical density* at 600 nm and NOT the absorbance.

Line 533: change on hundred microliters to 100 μL. (50mM Buffer should be (50mM buffer

Line 535: it is 750 μM NADPH and not 750 μM

Minor Changes Requested to Further Improve Manuscript

12. **Line 38:** Minor rewording suggested; change “Differently from” to “In contrast to”... to make the sentence: **In contrast to** the homomeric...

13. Minor: **Line 79:** Change the comma to a semicolon: “(tetrahydrosarcinapterin; *M. thermophila*) should be (tetrahydrosarcinapterin; *M. thermophila*)

14. **Line 156:** Change to either “...we obtain **a** more detailed insight” or “...we obtain more detailed insights”

15. **Line 180:** table 1 should be Table 1

REVIEWER COMMENTS

Reviewer #1 (Remarks to the Author):

In this manuscript, Filée and collaborators study the evolutionary history of folate metabolism in Asgard archaea. They particularly focus on the enzymes for thymidylate biosynthesis, for which they perform phylogenetic and genomic analyses, as well as functional characterization through expression in *E. coli* and biochemical analysis. The authors also expand their phylogenetic work on other enzymes that use folate.

This study contains a promising series of details and analyses. However, the general picture it recovers does not seem to provide a firm basis for the general themes that the authors outline, namely a novel understanding into eukaryogenesis, or broad insights on the ecology and evolution of Asgard archaea. The bioinformatic analyses are not conclusive and informative enough to accompany the author's conclusions. Finally, although the manuscript is generally well written, it contains many errors, unclear sentences and dubious arguments. I do, however, believe that the authors have made interesting progress on Asgard archaeal molecular biology, and I fully encourage them to keep building onto this topic.

We thank reviewer 1 for their encouraging words and interest in our work. We have improved the quality of the manuscript and removed some unclear or potentially dubious statements. We also stress that eukaryotic thymidylate synthase has a bacterial origin (this conclusion was not objected by the referees). Consequently, our study suggests that the capacity of eukaryotic cells to duplicate their genetic material is a sum of archaeal (replisome) and bacterial (thymidylate synthase) characteristics.

We have addressed the main points raised below.

1. Relevance and support to conclusions

The authors contextualize their study by touching some of the current major questions around Asgard archaeal biology. Indeed, Asgard archaea are fascinating organisms that are bringing much anew on our ideas of eukaryogenesis, and are thus attractive objects of study. However, this study does not seem to add on eukaryogenetic models, nor it adequately provides general insights onto Asgard archaeal ecology and evolution.

Eukaryogenesis. The authors do explicitly want to include evolutionary models of the evolution of thymidylate synthesis between the Asgard archaea and Eukaryotic groups, but very little progress is made on this front, other than suggesting a non-Asgard archaeal origin of ThyA and ThyX in eukaryotes. This question could be tackled much better. Moreover, I believe the authors should more explicitly explain why this is a specifically interesting question in the first place.

Although the scientific community agrees that the eukaryotic DNA replication machinery has an archaeal origin, it has remained enigmatic where metabolic enzymes, required for DNA precursor synthesis, in Eukarya come from. Our work indicates that thymidylate synthase, a unique marker for DNA precursor synthesis, has been frequently transferred from Bacteria to Asgard Archaea. In addition, in ThyA trees, eukaryotic thymidylate synthases group together with bacterial and not archaeal sequences (bootstrap support 100 for separation between the majority of archaeal and bacterial/eukaryal sequences, exemptions are halophiles). This provides strong support for the fact that during eukaryogenesis, the **metabolic capacity** to produce DNA originated in bacteria, and not in Asgard archaea. So, the capacity of eukaryotic cells to replicate their genetic material is a sum of archaeal and bacterial characteristics. This statement is of general interest in understanding the evolution of DNA itself as a genetic material.

General insights onto Asgard archaeal ecology and evolution. The authors include several general conclusions around this theme: (1) Asgard archaea contain plastic genomes with a large impact of horizontal transfer, (2) ThyA is ancestral while ThyX has been recently incorporated multiple times through horizontal transfer, (3) folate metabolism in Prometheoarchaeum includes a variety of folate-dependent enzymes that have been obtained in Asgard archaea from bacteria and may be key to syntrophic interactions. Unfortunately, none of these points seem to be strongly supported (see also below).

Considering the central role of C1 carriers in bacterial and archaeal metabolism, our results indicating that many folate-dependent enzymes in Asgard have bacterial origins are of great interest. We suggest that many Asgard archaea require the interspecies transfer of C1 donors for the synthesis of DNA, RNA, proteins, and some vitamins. In agreement with this notion, Imachi et al., (2020) added folic acid to their enrichment cultures, which may have contributed to the cultivation success of this species. We thus suggest that many Asgard archaea seemingly require external sources for their C1 metabolism.

The focus of our current work is on folate metabolism. Our phylogenetic analyses have indicated a sporadic phylogenetic distribution of folate-dependent enzymes and several HGT events between Bacteria and Asgard archaea (PurH, 2 transfers, bootstrap > 98; MTHFS, 2, > 87; MTHFR, 3, >88; Meth, 3, >97; FTHFS, 1, 100; FOLD, Thorarcheota groups with bacteria; Fola, several, > 87; for ThyX and ThyA see below). We have removed any references to genome plasticity as with the current information, it is difficult to justify any claims in this area.

2. Results: phylogenetic analyses

I have struggled to find a good way to provide feedback for several of the results. Particularly, the phylogenetic analyses are very difficult to interpret. They form the backbone of the present study, and yet the information provided is highly insufficient for adequate evaluation.

First, the followed Methodology does not provide all necessary details. Multiple key details are unclear, including how the authors downsample the initial dataset (phylum or genus?), how exactly the authors identify homologs in Asgard archaeal genomes, what the purpose of the reciprocal Blast search outside of Asgard archaeal sequences is, how the authors proceed once the final dataset is obtained (i.e. were the sequences aligned to the previous alignment, or re-aligned and trimmed de novo?). The authors then claim to have reconstructed maximum-likelihood trees using two different software, but they only show one tree per gene. The authors also claim to have performed “1000 bootstrapped data sets”, but it is unclear under exactly which conditions, as PhyML and IQ-Tree provide different approaches to this. More importantly, the authors need to clarify, for trees obtained with IQ-Tree, whether these replicates correspond to the “Ultrafast bootstrap” or the “Non-parametric bootstrap” algorithms. As an additional note, the use of “Smart model selection” as implemented in PhyML is also not state-of-the-art, as the models that are implemented in this method cannot deal with more complex, accurate models as the ones provided in IQ-Tree (e.g. Q.pfam, free-rate heterogeneity parameters, mixture models).

We have rewritten this part and now provided a more detailed description of the methodology that was used. We only show trees obtained with IQTree because the obtained results were superior in terms of statistical support to the results obtained using the older methods (but conclusions remained the same using the different methods).

Importantly, the trees are displayed in unclear formats. The authors provide rooted trees for most of their phylogenies: how were these roots selected? Additionally, providing support values as circle sizes is very poor practice, as this does not allow proper distinction of the corresponding values, which dramatically hampers the adequate interpretation of all supplementary phylogeny figures. This is particularly important in the possible case that the authors have used “Ultrafast bootstrap” values in IQ-Tree. It would also be helpful that the authors provide an adequate color legend for all tree figures, and that this is complete (what are the different shades of blue? What does yellow represent?)

We thank the reviewer for these thorough comments. All the trees we produced are unrooted trees to illustrate the relatedness of the leaf nodes, which does not require the ancestral root to be known or inferred. We have redrawn the trees in supplementary materials for reasons of visibility using the rectangular presentation (as suggested by referee 2) so they may appear as “pseudo rooted” but they are not. We have changed and confirmed color codes. Numeric bootstrap values have been indicated for the major branches in the supplementary figures. As the control, we have constructed phylogeny for MCM helicases. This is an excellent marker for the archaeal/eukaryotic DNA replication machinery as bacterial and archaeal/eukaryotic replicative helicases are non-orthologous. This robust tree indicates that all Asgard archaea form a monophyletic group with the other archaea (as for other replication genes such as B-type DNA polymerase or DNA Gyrase). Therefore, our methodology is capable of producing robust phylogenies even for a single gene phylogenesis.

Regarding interpretation (on top of the previous issues)), it would be helpful that the authors clarify why *Dictyostellium* and *Alvinella* are the only eukaryote sequence found in the ThyX tree. As far as we are aware, *Dictyostelium* is the only eukaryotic ThyX that has been experimentally confirmed. According to previous and current work, this results from HGT between bacteria and eukarya. All other potential ThyX sequences in databases may correspond to contaminating reads for bacterial symbionts of eukaryotes and were ignored (including *Alvinella*).

3. Results: horizontal gene transfer detection

The authors provide potentially interesting insights onto the horizontal gene transfer patterns of Prometheoarchaeum, which could help inform evolutionary models on evolutionary dynamics of Asgard archaea. However, the results provided from HGTector are not conclusive enough for sweeping statements. First, more information should be provided about the statistical robustness of the obtained candidate genes and, in its absence, deeper confirmatory analyses are needed (phylogenetic or synteny analyses are definitely good places to start). The authors also present a modified version of this software, but do not explain what those changes affect, thus requiring a reader to refer to Github to interpret results. Very importantly, the provided results can be severely compromised if we consider that the “Distal” comparisons may include eukaryotic sequences – which may be closer relatives to Asgard archaeal sequences than most archaeal sequences.

The starting point for our study was the detection of putative HGT transfers using the “BLAST best match method and *Psyn* ThyX”. However, it is well established that this only provides a rough first estimate for the phylogenetic history of genes and mainly detects recent HGT events. HGTector analyses perform all-against-all similarity searches and classify hits into “typical” and “atypical” gene pools using automatically selected criteria. The method has been validated using simulated genomic data and real datasets. We have confirmed the predicted transfers using more robust phylogenetic analyses and alternative methods for folate-dependent enzymes. One potential pitfall of our analyses is that the “self-group” consists only of *Psyn*. Indeed, when we repeated a similar analysis with

“*Thermococcus*” species where the self-group is much larger, we obtained a more scattered distribution for potential gene transfers (see below).

4. Results: synteny analyses and ancestry claims

These are a good complement to horizontal gene transfer detection. Moreover, these analyses are used by the authors as support to establish the evolutionary dynamics of ThyA and ThyX. The difference in synteny conservation between these two sets of homologs is apparent, however insufficient data is provided for this interpretation. First of all, these results are rarely placed together with phylogenetic results. For example, the authors claim rampant horizontal transfer of ThyX as derived from their lack of synteny, but what is the explanation for the lack of synteny between gene copies that indeed appear as closest relatives in the ThyX phylogeny in Fig. 4? (This is also made difficult by the apparent lack of congruency between the genomes displayed in Fig. 5C and Fig. 4: **where are the Hod, Borr Hermod and Odin sequences in Fig 4?**).

In our opinion, the lack of synteny cannot be used as supporting evidence for any evolutionary scenario. Hod, Borr, Hermod, and Odin sequences were removed because the corresponding contigs are very short (see materials and methods).

Importantly, the synteny results for ThyA do show a certain degree of gene neighborhood conservation, but this is purely restricted to groups of closely related genomes (e.g. no conserved synteny is detected between Thor and Hod, or between Thor and Loki). This may indicate vertical inheritance within these groups (are these genomes even representative of the entire groups?), but there is no indication of vertical inheritance before the divergence of these groups. Thus, the author's conclusion that ThyA was present in the ancestor of Asgard archaea is unsupported by these results or by the phylogenetic results in Fig. 4.

We agree that it is unclear at this time whether analyzed genomes are representative of the entire Asgard group. We have highlighted this in the text. We believe that the *thyA* synteny groups and phylogenetic analyses indicate that *thyA* was present in the ancestor of the analyzed Asgard. The major conclusions of our manuscript are independent of this conclusion.

Non-exhaustive minor comments: - Suppl. Fig 1: it would be useful to know how many of the top hits correspond to closely related genomes (i.e. true verticality signal).

This is a great idea and we thank the reviewer for the suggestion. Here is the table with an additional row reporting the proportion of Asgard hit on the top 100 BLASTP hits (see also supplementary Fig. 1):

Gene	First BLASTP Hit	E-value	% Archea Top100 Hit	% Asgard Top100 Hit
pls	Candidatus Lokiarchaeota archaeon	0.0	43	42
701	Candidatus Lokiarchaeota archaeon	2E-33	100	97
702	Candidatus Lokiarchaeota archaeon	6E-12	95	57
PyrL	Candidatus Lokiarchaeota archaeon	9E-74	100	97
704	Candidatus Lokiarchaeota archaeon	2E-32	11	8
705	Candidatus Lokiarchaeota archaeon	0.0	100	47
706	Candidatus Lokiarchaeota archaeon	1E-19	100	97
707	Candidatus Lokiarchaeota archaeon	2E-39	100	98
nikR	Candidatus Lokiarchaeota archaeon	2E-19	100	53
710	Candidatus Methanofastidiosa archaeon	7E-64	100	94
711	Thermoprotei archaeon	6E-21	100	21
thyX	Zixibacteria bacterium	2E-78	3	2
713	Lokiarchaeum sp. GC14_75	3E-43	94	94
714	Lokiarchaeum sp. GC14_75	1E-56	92	90
716	Nitrospiraceae bacterium	4E-87	34	11
717	Candidatus Lokiarchaeota archaeon	1E-47	87	44
pol	Candidatus Lokiarchaeota archaeon	0.0	77	47
radA	Candidatus Lokiarchaeota archaeon	5E-53	81	72
721	Candidatus Helarchaeota archaeon	1E-45	39	12

This result indicates that an important proportion of the hits come from Asgard which support the idea that these genes have been inherited vertically and/or horizontally transferred between Asgards.

It would be interesting to further discuss the relevance of the patchiness of these genes in Asgard archaea. What does this mean for their biosynthetic capabilities, syntrophy potential and in general the evolution of their metabolism? Very importantly, what is the meaning of the substantially larger representation of these genes in *Promethoarchaeum*, even in comparison to its closest relatives? Do these results indicate that the folate-dependent metabolic networks in Asgard archaea are generally quite limited, given the predominance of gene absences rather than the opposite?

We have addressed this further in the discussion. Our work has raised the possibility that Asgard archaea are dependent on the transfer of C1 donors from their environment or symbiotic partners, similar to mammalian cells that cannot synthesize folates *de novo*. Sporadic acquisition of folate-dependent enzymes may have contributed to the fitness of Asgard archaea under different environmental conditions. One way to test this would be to add the different folate derivatives to the growth medium of Asgard archaea; however, only few labs in the world are capable of performing these experiments.

Reviewer #2 (Remarks to the Author):

In this paper titled "Bacterial Origins of Thymidylate and Folate Metabolism in Asgard Archaea" the authors Filée et al., have performed a detailed and comprehensive analysis of the folate metabolism in Asgard archaea. This newly discovered group has an interesting placement within the context of eukaryogenesis and hence has garnered much interest although the present analysis does not pertain to this but rather microbial biochemistry which is refreshing. I only have some minor suggestions which the authors can consider.

Did the authors consider performing Gene-tree to species tree reconciliations maybe using ALE in order to further dissect the source and history of transfers in the ThyX and other genes? This could help

resolve some uncertainty in the conclusions that have resulted from HGTector. It is potentially easy to check and only needs a species tree which should be easily constructible.

We have performed the suggested analyses for *thyX* and *thyA* using a 16S rDNA tree as species tree and the parsimonious reconciliation analyses using the ecceTERA software. This result provides further support for our conclusions with two transfers between Asgard and bacteria for ThyX, none for ThyA (and several intra-archaea HGTs for both). However, these reconciliation analyses are based on a subset of the initial data set, as many Asgard MAGs lack complete 16S rDNA sequences and were removed from the analysis. Thus, this estimation should be considered as a minimal number. These reconciliations also have their limits. As it is difficult provide reliable time-trees for deep-rooted phylogenies, the direction and the timing of the transfers remain elusive. Nevertheless, we have included a new supplementary figure with the two reconciliation trees that support our conclusions. We discuss these results in the manuscript.

The authors mention the study by Garg et al, to say that they (Garg et al.) concluded that Asgard MAGs display evidence of assemble errors but as far as I understand they only looked at ribosomal proteins which may cause an issue when used in a concatenated alignment. This would be important for the authors when they do indeed do the reconciliation analyses. However, these should be mitigated by the inclusion of high-quality genomes. I would advise the authors to not misrepresent studies and inflate or deflate claims in a brad stroke.

We have modulated this statement and apologize for our mistake.

The phylogenetic trees appear to be of a low quality and a little difficult to read even in the excellent ppt of the figures provided. Maybe these should be checked or re-colored before publication to be color blind friendly. Also, rectangular representations tend to be more accurate and easier than circular notations.

These have been now redrawn using a rectangular presentation as suggested. We stress that all trees indicated in this manuscript are unrooted (see our comment above). Some trees in a supplementary material are not visible if printed in the A4 format; however, their electronic format is of high quality and can be zoomed in.

In summary, I would like to see the results of a reconciliation analyses which would enrich what is already an extremely competent study. Thank you

Thank you for your constructive comments.

Referee 3:

The article entitled “Bacterial Origins of Thymidylate and Folate Metabolism in Asgard Archaea” by Filée et al describes an extensive bioinformatic search for an archaeal flavin dependent thymidylate synthase (FDTS) and both the identification, purification and partial biochemical characterization of the enzyme. This discovery is significant and represents the first demonstration of an archaeal FDTS which has important evolutionary implications. The recommendation of the reviewer is to **publish after minor revisions** that are noted below. Before providing my suggested improvements, I must disclose that I am a mechanisticenzymologists whose training is in the fields of protein biophysics/biochemistry and bioorganic chemistry. So I was unable to provide a critical review of the bioinformatic and

phylogenetic analysis other than to say the authors should be commended for describing these parts in a manner that a non-expert can easily follow. I also am comfortable recommending publication because the authors' analysis clearly found an archaeal FDTS. The enzymology was sound and the authors did *not* overstate the kinetic data as I often see without the proper controls.

We thank the referee for their helpful comments.

Substantial but still Minor Changes Needed

1. **Figure 1 and throughout the text:** *Enzymes* carry out each of the reactions mentioned not the genes that encode for them. ThyA, FoaA and GlyA should be changed to thymidylate synthase (TS), dihydrofolate reductase (DHFR) and serine hydroxymethyl transferase (SHMT), respectively. Listing the name of the enzymes is more customary for both biochemists and biophysicists. It was distracting to constantly read the names of the genes especially for the thymidylate synthases. Call the classical one TS or TSase and the flavin dependent one FDTS or FDTSase (the reviewer suggests leaving off "ase" for simplicity).

This is significant to improve the scientific rigor of the article but an easy edit to make. The *thyX* gene for example is expressed regardless of if FAD is present or not. Also, ThyX-FAD is written in Scheme 1 over the reaction arrows. This gives the impression that the flavin acts as a cosubstrate and that binds, reacts and is released in a manner similar to the nicotinamide. This is not the case; FAD is non-covalently bound but remains associated to the enzyme throughout the catalytic cycle as is common in flavoenzymes.

The text will need to be fixed throughout the manuscript to more clearly denote the enzymes when they are the molecules being discussed and to not confuse them with the genes.

Thank you for these comments. We have modified Fig. 1 and removed FAD from the ThyX reaction scheme and agree with the referee.

We are using bacterial nomenclature for genes and proteins. For instance, this means that *thyX* refers to genes and **ThyX** refers to protein. We have also introduced FDTS as an alternative name. This may reflect the differences between the different domains of science (microbiology, genetics, chemistry).

2. **Line 36:** Two additional references should be included for FDTS discovery and mechanism as listed:

a. "Mishanina, T.V., Yu, L., Karunaratne, K., Mondal, D., Corcoran, J.M., Choi, M.A., Kohen, A., *Science*, **2016**, 351, 507-510."

b. "Koehn, E.M., Fleischmann, T., Conrad, J.A., Palfey, B.A., Lesley, S.A., Mathews, I.I., and Kohen, A.*, *Nature* **2009**, 458, 919-9233

3. **Lines 39-41:** This requires editing since it is confusing as written. "...accommodates dUMP, NADPH, and the carbon donor methylene tetrahydrofolate (CH₂H₄folate, a vitamin B9 derivative),

These important references have been added as suggested.

is located at the interface of three..."

Change to: "...accommodates dUMP, NADPH, and the carbon donor methylene tetrahydrofolate (CH₂H₄folate, a vitamin B9 derivative). CH₂H₄folate is located at the interface of three subunits..."

It has been six or seven years since I have studied the structure of FDTS so am unsure if the authors intend to say that CH₂H₄folate is at the interface or that all three substrates (CH₂H₄folate, dUMP and folate) are at the interface. Please clarify.

We have modified the sentence to clarify its meaning. This sentence refers to our earlier mutagenesis studies that indicated that amino acid residues from three monomers are forming a single active site of the ThyX enzyme (Leduc et al. PNAS 2004).

4. **Line 55:** “The ThyX reaction mechanism **appears** less catalytically efficien...” FDTS is known to be less efficient since it suffers from significant substrate inhibition. While the k_{cat}/K_m values are roughly similar the significant substrate inhibition makes FDTS a more sluggish enzyme. Since it would distract from the impressive work here to cite the kinetic papers the authors should simply remove appears and state: “The **FDTS** reaction mechanism **is** less catalytically efficient...”

We have modified the sentence and added the reference for substrate inhibition. We earlier reported that k_{cat}/K_m values of ThyX are approximately 10% of those reported for ThyA.

5. **Line 74:** Perhaps this is because I am not a molecular biologist but this completely unclear to me. What exactly was done in reference 3? I assume (did not check the ref at all) they knocked out the gene and visualized on a gel? Functional *thyX* to me means that FDTS was expressed and was able to make dTMP. A few words on what is meant by genetic evidence would help to clarify.

Genetic evidence refers to complementation tests. We have slightly modified the text to indicate this.

7. I think but am not certain *Promethoarchaeum syntrophicum* should be in italics. Please check and correct if needed.

We have not indicated “*Promethoarchaeum syntrophicum*” in italics because it has still a “Candidatus” status. Our understanding is that this is a current naming convention.

8. **Line 134:** This is not my area of expertise but the casing must surely be wrong here. If so please correct. If not please excuse my ignorance: “Reads Per Kilobase of transcript, per Million mapped reads”.

The use of capital letters refers to a common abbreviation “RPKM”.

9. **Line 189 entire section:** I must admit as a mechanistic enzymologist this is entirely out of my field. I read through all of the genomic data and was able to clearly see what the authors did and were presenting but have no expertise to judge the merit of the work.

We have addressed comments from referees 1 and 2 above in this area.

10. ***Psyn thyX* is functional in *Escherichia coli*:** The description of the plasmid is too long especially considering the plasmid map is shown in Figure 6a. The authors simply need to note it has a *lac* promoter, the gene to express the his tagged protein and that more details are shown in

Figure 6a. This does not require 8 lines of text.
The text was shortened as suggested.

Also the termed “scored” after 3 days suggests a quantitative measure of cell growth was taken. This was not done and is not needed to make the authors point. “Scored” should therefore be changed to “checked”. This should also be changed on **line 507** in the methods.
Scored was changed to checked as suggested.

Finally, please note in the text that both plates shown were grown 37 °C. I wondered if the additional growth observed on the plate to the right was due to temperature or the addition of thy. It was listed in the Figure legend but I did not see it in the text and this is important to stress the point the authors demonstrated.
The residual growth is due to the intracellular stock of thymidine found in cells. This is the reason why we checked the formation of individual colonies in the absence of thymidine.

11. The material and methods needs substernal proofreading. I am only going to list a few example here. The most obvious is that a space must be indeted after the number and before the unit. For example 50 mM and not 50mM. I noticed this in the results too and it should be fixed. It became annoying in the methods because the authors were not consistent and sometimes even spelled out the number and the unit.

We have rechecked these points and apologize for any remaining mistakes.

Line 481: typo. environments is misspelled.

Line 508: IPTG needs to be spelled out as isopropyl β-D-1-thiogalopyrnoside the first time it is mentioned. Delete (shown in Fig 5) this belongs in the results not the methods.

Line 513: light is not being absorbed but instead one is monitoring growth by looking at the turbidity of the media. Therefore, it is the *optical density* at 600 nm and NOT the absorbance.

Line 533: change on hundred microliters to 100 μL. (50mM Buffer should be (50mM buffer

Line 535: it is 750 μM NADPH and not 750 μM

Minor Changes Requested to Further Improve Manuscript

We have made these corrections and changes in the manuscript as suggested.

12. **Line 38:** Minor rewording suggested; change “Differently from” to “In contrast to”... to make the sentence: **In contrast to** the homomeric...

13. Minor: **Line 79:** Change the comma to a semicolon: “(tetrahydrosarcinapterin; *M. thermophila*) should be (tetrahydrosarcinapterin; *M. thermophila*)

14. **Line 156:** Change to either “...we obtain **a** more detailed insight” or “...we obtain more detailed insights”

15. **Line 180:** table 1 should be Table 1

Reviewer #2 (Remarks to the Author):

I thank the authors for incorporating the comments suggested by the reviewers. R1 indeed has raised some valid concerns which I think the authors have addressed to the best of their ability and they remain satisfactory.